# Did Holocene climate changes drive West Antarctic grounding line retreat and readvance?

Sarah U. Neuhaus[1], Slawek M. Tulaczyk[1], Nathan D. Stansell[2], Jason J. Coenen[2], Reed P. Scherer[2], Jill A. Mikucki[3], Ross D. Powell[2]

[1]Earth and Planetary Sciences, University of California Santa Cruz, Santa Cruz, CA, 95064, USA
[2]Department of Geology and Environmental Geosciences, Northern Illinois University, DeKalb, IL, 60115, USA
[3]Department of Microbiology, University of Tennessee Knoxville, Knoxville, TN, 37996, USA

*Correspondence to*: Sarah U. Neuhaus (suneuhau@ucsc.edu)

**Abstract.** Knowledge of past ice sheet configurations is useful for informing projections of future ice sheet dynamics and for calibrating ice sheet models. The topology of grounding line retreat in the Ross Sea Sector of Antarctica has been much debated, but it has generally been assumed that the modern ice sheet is as small as it has been for more than 100,000 years (Conway et al., 1999; Lee et al., 2017; Lowry et al., 2019; McKay et al., 2016; Scherer et al., 1998). Recent findings suggest that the West Antarctic Ice Sheet (WAIS) grounding line retreated beyond its current location earlier in the Holocene and subsequently readvanced to reach its modern position (Bradley et al., 2015; Kingslake et al., 2018). Here, we further constrain the post-LGM grounding line retreat and readvance in the Ross Sea Sector using a two-phase model of radiocarbon input and decay in subglacial sediments from six sub-ice sampling locations. In addition, we reinterpret high basal temperature gradients, measured previously at three sites in this region (Engelhardt, 2004), which we explain as resulting from recent ice shelf re-grounding accompanying grounding line readvance. At one location – Whillans Subglacial Lake (SLW) – for which a sediment porewater chemistry profile is known, we estimate the grounding line readvance by simulating ionic diffusion. Collectively, our analyses indicate that the grounding line retreated over SLW $4300^{+1500}_{-2500}$ years ago, and over sites on Whillans Ice Stream (WIS), Kamb Ice Stream (KIS), and Bindschadler Ice Stream (BIS) $4700^{+1500}_{-2300}$, $1800^{+2700}_{-700}$, and $1700^{+2800}_{-600}$ years ago respectively. The grounding line only recently readvanced back over those sites $1100^{+200}_{-100}$, $1500^{+500}_{-200}$, $1000^{+200}_{-300}$, and $800 \pm 100$ years ago for SLW, WIS, KIS, and BIS respectively. The timing of grounding line retreat coincided with a warm period in the mid- to late-Holocene. Conversely, grounding line readvance is coincident with cooling climate in the last 1000-2000 years. Our estimates for the timing of grounding line retreat and readvance are also consistent with relatively low carbon-to-nitrogen ratios measured in our subglacial sediment samples (suggesting a marine source of organic matter) and with the lack of grounding-zone wedges in front of modern grounding lines. Based on these results, we propose that the Siple Coast grounding line motions in the mid- to late-Holocene were

primarily driven by relatively modest changes in regional climate, rather than by ice sheet dynamics and glacioisostatic rebound, as hypothesized previously (Kingslake et al., 2018).

## 1 Introduction

Ice loss from the West Antarctic Ice Sheet (WAIS) is a significant uncertainty in projections of near-future sea-level rise (Bamber et al., 2019; Church et al., 2013). This uncertainty is partly due to limited observational constraints on the climate sensitivity of WAIS grounding lines, which mark the locations where ice thins and starts floating on seawater. The evolution of grounding line positions in West Antarctica during and immediately following the Last Glacial Maximum (LGM) is well documented because the relevant geologic evidence is largely accessible to marine geophysical and geological investigations (Anderson et al., 2014). However, evidence of WAIS grounding lines during much of the Holocene is hidden underneath fringing ice shelves, or even beneath the ice sheet itself (e.g., Bradley et al., 2015; Kingslake et al., 2018; Smith et al., 2019; Venturelli et al., 2020). Early ideas about post-LGM grounding line retreat in WAIS were that the grounding lines retreated unidirectionally, and that the modern configuration is the smallest since the LGM (Ackert, 2008; Anderson et al., 2014; Bentley et al., 2014; Conway et al., 1999; Hall et al., 2013). However, recent studies have challenged this paradigm by suggesting that during the Holocene, at least some of WAIS grounding lines retreated behind their modern positions before readvancing to their current location (Bradley et al., 2015; Kingslake et al., 2018; Matsuoka et al., 2015; Venturelli et al., 2020). Additionally, exact timings and the manner in which the grounding line retreated have been much debated (Ackert, 2008; Bart et al., 2018; Conway et al., 1999; Halberstadt et al., 2016; Kingslake et al., 2018; Lee et al., 2017; Lowry et al., 2019; McKay et al., 2016; Prothro et al., 2020; Spector et al., 2017).

Understanding Holocene ice sheet behavior may aid current efforts to project near-future ice mass loss from WAIS, due to the similarity of Holocene climatic and glacioisostatic forcings to the modern and near-future conditions. Here we re-examine previously collected datasets and samples to estimate the timing of grounding line retreat from, and readvance to, the modern configuration in the Ross Sea sector of WAIS. We show that the available evidence is consistent with a climatic forcing of these major grounding line movements, with the retreat occurring during warm periods of the Holocene (Cuffey et al., 2016), which are also characterized by a decrease in sea-ice cover in the Ross Sea (Hall et al., 2006). Readvance occurred in the last 2000 years, corresponding to cooling recorded in the WAIS Divide ice core and an increase in Ross Sea summertime sea ice (Cuffey et al., 2016; Hall et al., 2006). Ice sheet model results published in Kingslake et al. (2018) suggest that these changes in Holocene positions of WAIS grounding lines may have been associated with global sea-level variations of about 0.2-0.3 m at the time when temperature variations at the WAIS Divide ice core site amounted to just a few degrees (Cuffey et al., 2016; Cuffey, 2017). Previous research in Greenland has indicated that at least some of the Greenland Ice Sheet retreated behind its present margin during the mid-Holocene climatic optimum; simulation models suggest this loss of ice during retreat contributed an equivalent of 0.1-0.3 m of global sea-level rise (Vasskog et al., 2015).

## 2 Methods

### 2.1 Modelling of Basal Temperature Gradients

Vertical temperature profiles through ice can reveal information about past climatic conditions or ice behavior. The work of Engelhardt (2004) highlighted the fact that a large fraction of vertical temperature profiles measured in the Siple Coast ice streams are difficult to explain with an assumption of climatic and ice dynamic steady state because observed temperature gradients in the lowermost 100 m are anomalously high. The high basal temperature gradients indicate that there is cold ice present near the base of the ice. There are three ways to get cold ice at the base of an ice sheet without changing the basal boundary conditions: cooling the surface temperature, increasing the vertical advection (accumulation), or increasing the horizontal advection (ice velocity). Engelhardt (2004) addressed these options and ruled out increased accumulation or lowering of the surface temperature, ultimately settling on horizontal advection as the main explanation for the cold ice. However, even by proposing a "super-surge" event that drastically increased the horizontal advection, he was unable to reproduce the high basal temperature gradients without invoking basal melt (i.e. changing the basal boundary conditions).

Here, we conjecture that this observed unsteady thermal state is due to changing basal boundary conditions from Holocene ice shelf re-grounding in the region (Kingslake et al., 2018; Venturelli et al., 2020). Thus, we modelled the evolution of basal temperature gradients to constrain the timing of ice shelf grounding consistent with observed high basal temperature gradients. Measurements and analysis of temperature profiles taken at our field sites have previously been published (Engelhardt, 2004; Engelhardt and Kamb, 1993; Kamb, 2001); here we re-analyzed these profiles with the new assumption that this area (i.e., the Siple Coast) was an ice shelf that grounded in the recent past. We focused on the sites that had very steep basal temperature gradients: Kamb Ice Stream (KIS), Bindschadler Ice Stream (BIS), and the Unicorn (UC) (Fig. 1) (Table 1). (N.B. In this paper we distinguish between our field sites and the ice streams they are located on by using abbreviations to refer to the field sites, and full names to refer to the ice streams.) Building on modelling employed by Bindschadler et al. (1990) to date the formation of Crary Ice Rise as the Ross Ice Shelf grounded on a bathymetric high, we modelled temperature profiles of an ice shelf before and after grounding. We then compared the modelled basal temperature gradients to observed basal temperature gradients (Engelhardt, 2004). Our MATLAB code solves a one-dimensional, forward Euler, vertical advection-diffusion equation with a one-year time step and 10 m vertical step chosen to satisfy the von Neumann stability condition. The surface accumulation rate is set to be equal to the sum of vertical advection and change in ice thickness. Thus, when ice thickness remains constant, vertical advection is equal to the accumulation.

We ran the ice temperature model in two phases: floating ice shelf and grounded ice. First, we ran a diffusion advection model for the ice shelf phase with a constant surface temperature of -25 ºC based on surface temperature measurements along the Siple Coast (Engelhardt, 2004) and a bottom temperature equal to the freezing point of seawater at a

salinity of 34 PSU and a pressure corresponding to ice thickness, calculated using Eq. (5) in Begeman et al. (2018). We assumed a constant surface accumulation rate of 0.15 m/yr based on observations at Siple Dome (Waddington et al., 2005). Additionally, we assumed the ice shelf to be in steady state, which requires a basal melt rate of 0.15 m/yr to compensate for accumulation, and ignored horizontal advection. We varied the starting ice shelf thickness from 500 m to 1000 m based on modern ice thickness near the Ross Ice Shelf grounding lines (Fretwell et al., 2013; Still et al., 2019) and allowed the ice shelf temperature profile to come to steady state. Using this ice shelf steady-state temperature profile as the initial conditions for the second phase of the simulations, we then modelled the ice temperature evolution after grounding. Keeping the surface boundary conditions constant, we changed the bottom temperature boundary condition to reflect the pressure melting-point of ice. Although other similar models – including the Bindschadler et al. (1990) model – pick the basal boundary condition after grounding to be the geothermal flux, we chose the freezing point of freshwater because we assumed that basal freeze-on occurred at these locations after grounding, as evidenced by the widespread presence of basal ice layers found in boreholes drilled along the Siple Coast ice streams (Christoffersen et al., 2010; Kamb, 2001; Vogel et al., 2005) and at the grounding zone of the Whillans Ice Stream (unpublished data). Because basal melt rates of ice shelves near grounding lines are high – on the order of 10 cm/yr (Begeman et al., 2018) or even 20 m/yr in melt channels (Marsh et al., 2016) – we assume that any basal ice present prior to grounding line retreat melted away when our field sites became ungrounded, and that the basal ice found along the Siple Coast formed after re-grounding. In addition, we calculated the thickness of basal ice that forms during Phase 2 of the model. We allowed the model to run from 0 to 8000 years, and obtained the time-dependent temperature gradient for the bottom 100 m so we could compare our modelled results to observed basal temperature gradients (cf. Engelhardt, 2004). We chose 8000 years ago as the earliest the grounding line could have retreated over our field sites based on grounding line positions inferred from the dating of sediments in the Transantarctic Mountains and the Ross Sea, which placed the grounding line several kilometers north of the Siple Coast ice streams 8000 years ago (Lee et al., 2017; McKay et al., 2016; Spector et al., 2017). Sensitivity tests and equations used for our ice temperature model are presented in the Supplemental Material.

## 2.2 Ionic Diffusion Modelling

To estimate the timing of retreat and readvance of the grounding line at Whillans Subglacial Lake (SLW) (Fig. 1), where the observed basal temperature gradients are not anomalously steep (Fisher et al., 2015), we compared measured porewater ionic concentrations from a sediment core collected at SLW (Michaud et al., 2016a) to values modelled using an ionic diffusion model with a two-stage upper boundary condition. The first phase here assumes that the sedimentary column at SLW was exposed to seawater for some length of time during the Holocene ($T_o$). This was followed by ice-shelf re-grounding and exposure of the SLW sedimentary column to basal meltwater ($T_i$). Porewater chemistry data that we compare to the output of our forward model comes from sediment core MC-3B collected from SLW using a multicorer on January 30, 2013; methods used for sediment core and porewater collection are described in detail in Tulaczyk et al. (2014) and Michaud et al. (2016b).

We ran one-dimensional vertical diffusion simulations using the chemical parameters Cl⁻, SO₄²⁻, Na⁺, Ca²⁺, $\partial^{18}O$, and $\partial D$, through pore spaces of sediments below SLW (cf. Adkins and Schrag, 2003). The diffusion coefficients used for each chemical parameter examined in this study were calculated using the equation from Li and Gregory (1974) for diffusion of ions through sediments:

$$D_{sed} = D \frac{\alpha}{\theta^2} \tag{1}$$

Where $D$ is the diffusion coefficient in bulk water, $\alpha$ is the ratio of viscosity of the bulk solution to the average viscosity of the interstitial solution and $\theta$ is tortuosity. $\alpha$ is a constant with value close to one (Li and Gregory, 1974). In this study we assume it to be exactly one for simplicity. We used previously calculated bulk-water diffusion coefficients determined for the different chemical species at 0 ℃ (Li and Gregory, 1974; Wang, 1951a, 1951b). To estimate tortuosity, we use an equation from Boudreau (1996):

$$\theta^2 = 1 - \ln(\varphi^2) \tag{2}$$

Where $\varphi$ represents porosity, which is taken to be 0.4 here based on previous observations of till porosity along the Siple Coast (Engelhardt et al., 1990; Tulaczyk et al., 2001). The final diffusion coefficients used for each chemical parameter are shown in Table (2).

We forced the diffusion model by switching the upper boundary conditions to reflect either marine or subglacial conditions (Table 2), which represent time periods when the grounding line had retreated beyond or advanced over SLW, respectively. For modelling purposes, we assumed the switch between sub-ice shelf and subglacial conditions and vice versa was instantaneous. However, we acknowledge that the transition is probably more nuanced and includes a stage of estuarine conditions where tidal pumping allows marine waters to enter the subglacial system upstream of the grounding line, as seen currently under the Whillans Ice Stream (Horgan et al., 2013a). The depth of the simulated diffusion profile was 100 m, vertical resolution was 0.5 m, and temporal resolution was 1 year. We started each simulation assuming that the initial porewater through the entire profile was in equilibrium with subglacial meltwater conditions, i.e., meltwater chemical properties of Whillans Subglacial Lake (Table 2) throughout the domain thickness. This assumption is justified by the fact that the SLW site was beneath the ice sheet for at least 30,000 years (Clark et al., 2009) and must have been in a subglacial setting for almost all of the Quaternary period when WAIS was mostly larger than today (e.g., Bentley et al., 2014). We then instantaneously changed the upper boundary condition at time $t = 0$ to reflect seawater values (Table 2) to represent post-LGM grounding line retreat past SLW. In this first phase, characterized by simple initial and boundary conditions, the time-

and depth-dependent changes in chemical concentration for each parameter can be expressed using the analytical solution (Turcotte and Schubert, 2014, Eq. 4.113):

$$\frac{C - C_o}{C_s - C_o} = erfc\left(\frac{y}{2 * \sqrt{D_{sed} * t}}\right)$$ (3)

Where $C$ is the ion or isotope concentration, subscripts $o$ and $s$ represent initial and surface respectively, $y$ is the distance from the surface, and $t$ is the total time. We then instantaneously changed the boundary conditions at time $t = T_o$ back to subglacial meltwater conditions to simulate ice shelf re-grounding. At $t = T_o$ we also switch to solving the problem using a finite-difference diffusion model, which is a modified version of the same MATLAB code used by us previously to calculate

vertical heat advection-diffusion in the ice shelf. The initial condition for the second, numerical phase of the model is the profile obtained from our analytical solution (Eq. 3) at time $t = T_o$. The subglacial and oceanic tracer concentrations we assumed for the six chemical parameters are given in Table (2). We ran the two-phase analytical-numerical diffusion model for each chemical parameter separately, varying the amount of time during which the upper boundary condition reflected marine conditions and subglacial conditions ($T_o$ and $T_i$) from 1 to 8000 years.

To determine which profiles fit the measured concentrations in the core, we compared the gradient of the top 0.5 m of each model run to the measured concentration profiles (Michaud et al., 2016a). We performed a least-squares regression on the measured concentrations and tracked which model profiles fit within 95% confidence bounds for the measured values.

We also considered the possibility that vertical advection was occurring as groundwater flow, but were able to discount it based on considerations of the Peclet number:

$$Pe = \frac{uH}{D_{sed}},$$ (4)

where $H$ is the length of the core and $u$ is groundwater velocity. In order for advection to be dominant (i.e. $Pe \geq 10$), $u$ must be at least 0.011 m/yr. However, this is two orders of magnitude greater than previous calculations of upward groundwater flux on the Whillans Ice Stream (Christoffersen et al., 2014).

**2.3 Radiocarbon Measurements and Modelling**

We re-evaluated the radiocarbon data from the Siple Coast of the WAIS presented in Kingslake et al. (2018) to estimate the timing of Holocene grounding line retreat and readvance in this region. We focused on 11 subglacial till samples collected between 1989 and 2013, and 23 sediment samples collected below the Ross Ice Shelf in 1978 and 2015. Site locations are shown in Fig. 1 and include: SLW, the Whillans Ice Stream (WIS), KIS, BIS, the Whillans Grounding Zone (WGZ), and the Ross Ice Shelf Project (RISP). Details of the core collection, storage, and radiocarbon analysis are

described in Kingslake et al. (2018). Apparent carbon ages inferred from the fraction of modern radiocarbon ($Fm$) in the samples were presented in the extended data section of Kingslake et al. (2018). Kingslake et al. (2018) were careful to note that the apparent radiocarbon ages were not true ages due to the fact that the samples consisted of a mixture of young

radiocarbon-bearing organic matter and reworked radiocarbon-dead organic matter, which resulted in carbon ages that are much older than the actual age of the last exposure to radiocarbon inputs.

To better understand the sources of organic matter found in our sediment samples, we collected new data on carbon and nitrogen present in the samples: 17 samples collected from subglacial sites, 17 samples from sub-ice shelf sites, and 2 samples melted out from basal ice collected at a subglacial site. Total carbon (TC), total organic carbon (TOC), and total nitrogen (TN) measurements were performed at the University of California Santa Cruz Stable Isotope Laboratory. Samples (8-10 mg bulk sediment) for TOC and $\delta^{13}C$ determination were decarbonated via direct acidification with sulfurous acid. All

samples were dried prior to weighing for measurement. TC and TN were measured simultaneously by Dumas combustion using a CE Instruments NC2500 elemental analyzer coupled to a ThermoFinnigan Delta Plus XP isotope ratio mass spectrometer. TOC and $\delta^{13}C$ were measured independently on the same instrument. All measurements were calculated relative to an in-house gelatin standard, which was extensively calibrated against international standard reference materials. Typical reproducibility ($1\sigma$) of duplicate measurements is better than 0.1 wt%C, better than 0.01 wt%N, and better than 0.1

permil $\delta^{13}C$. Finally, total inorganic carbon (TIC) was calculated as the difference of TC and TOC. The TOC and C:N measurements for our field sites are shown in Fig. 2, and all results for TOC, TC, TIC, C:N, and $\delta^{13}C$ are presented in the Supplemental Material. We used TOC measurements to constrain our radiocarbon model (see below) and used $\delta^{13}C$ and C:N measurements to better understand the origin of the sediments. By examining where the measurements lie on a $\delta^{13}C$ vs. C:N plot (e.g., Fig. 3) we can glean information about the potential sources of the organic material in the sediments (Lamb et

al., 2006).

         To estimate the timing of retreat and readvance of the grounding line along the Whillans Ice Plain, we developed a two-phase model of $^{14}C$ and $^{12}C$ evolution at our field sites from 8000 years ago to the present (see Fig. 4 schematic). We again used 8000 years ago as the earliest possible time the grounding line could have retreated behind our field sites. The first model phase represented the time after grounding line retreat beyond the sediment sampling locations (Fig. 4). We

assumed that at the onset of post-LGM grounding line retreat in the Ross Sea Sector, sediments at our field sites contained no $^{14}C$, but did contain organic matter with $^{12}C$ (Fig. 4). Given the short half-life of radiocarbon and the geologic evidence suggesting that the grounding line in the Ross Sea was at its LGM maximum position at roughly 30,000 years ago, we followed the conjecture of Kingslake et al. (2018) that $^{14}C$ found in the sediments was incorporated after the post-LGM grounding line retreat. This assumption that the $^{14}C$ originated from a marine environment was also employed by Venturelli

et al. (2020) when they dated grounding line retreat over WGZ. New, $^{14}C$-bearing marine organic matter was thus introduced into radiocarbon-free sediments during this first model phase (Fig. 4). We assumed a constant rate of $^{14}C$ deposition, $a$, and a constant rate of $^{12}C$ deposition, $A$, for the entire first phase of the model, when sediments were assumed to be exposed to input of radiocarbon-bearing marine organic matter. The ratio of $^{14}C$ to $^{12}C$ at the time of deposition was taken to be equal to that measured in modern amphipods collected at the grounding zone of the Whillans Ice Stream (WGZ)

in 2015 (Kingslake et al., 2018). Because these amphipods were part of the marine food chain beneath the Ross Ice Shelf,

we assumed that their *Fm* is representative of the *Fm* of the ocean water in the grounding line environment. The *Fm* for the amphipods corresponds to a radiocarbon reservoir age of ca. 1000 years for the sub-ice shelf marine organic matter. We represented the evolution of $^{14}$C concentration, *n*, during this phase by accounting for both the addition and decay of $^{14}$C, with the equation:

$$n(t) = a\tau \left(1 - e^{\frac{-t}{\tau}}\right) \tag{5}$$

Where *t* is time and $\tau$ is the mean lifetime of $^{14}$C (8033 years, based off the Libby half-life of $^{14}$C [Stuiver and Polach, 1977]). We represented the time-dependent concentration of $^{12}$C (*N*) during the first model phase using the equation:

$$N(t) = N_o + At \tag{6}$$

Where $N_o$ is the amount of $^{12}$C present initially in the system. We ran the first phase of the model from $t = 0$ to $t = T_o$, where $T_o$ is the length of time a given field site is exposed to the ocean. The second model phase represents the time after the ice
sheet again covered a given site, and the only process affecting $^{14}$C concentration is radioactive decay (Fig. 4). We represented the $^{14}$C concentration with the equation:

$$n(t) = n^* e^{\frac{-t}{\tau}} \tag{7}$$

Where $n^*$ is the value of *n* when the system switches from sub-ice shelf to subglacial. After a site is glaciated, inputs of marine organic matter cease, and thus we took $^{12}$C to be constant with time at the value it had at the moment of grounding $(N^*)$:

$$N(t) = N^* \tag{8}$$

Detailed derivation of these equations is shown in the Supplemental Material.

We normalized the calculated TOC values to a 100 g sample of dry sediments. We added *n(t)* and *N(t)* and divided by 100 g of sediment. To calculate *Fm*, we divided *n(t)* by *N(t)*, and divided the quotient by the modern ratio of $^{14}$C to $^{12}$C
$(1.176 \times 10^{-12})$.

We assume that organic carbon input to the sediments comes predominantly from local fecal pellets and necromass of macro, micro, and meiofauna, with a potential additional syndepositional regional input by sub-ice shelf water column advection (Kingslake et al., 2018; Turner, 2015). During borehole drilling at WGZ, where the water column was only 10 m

thick, higher than expected concentration of living biomass were observed in the form of planktic and nektonic organisms, including amphipods, fish, and jellies (whereas no multicellular life was observed at SLW). The only potential evidence noted of benthic organisms at WGZ was rare organic sheaths, presumably from infaunal meiofauna at the sediment-water interface. The absence of a developed benthic community is likely due to high sediment rainout from melt of debris-laden basal ice, as observed by borehole cameras. Low concentrations of infauna were noted in the upper 10 cm at RISP (Barrett, 1975; Harwood et al., 1989; Kellogg and Kellogg, 1981), ca. 100 km from WGZ, which is reflected in the relatively higher $Fm$ measured in those samples.

Because it is difficult to discern a sediment flux rate at WGZ from these observations, we determined the accumulation of carbon in the sub-ice shelf ocean cavity by running the model for both RISP and WGZ, where we could ignore the readvance of the grounding line (phase two of the model). Here, we expect them to be similar to the sub-ice shelf conditions experienced by our field sites following grounding line retreat, i.e., in the first phase of the model. We thus ran phase one of the model for RISP and WGZ for $t$ equaling $8000 \pm 1000$ years. Because previous studies have placed the grounding line at Ross Island ca. 8000 years ago (Baroni and Hall, 2004; Licht et al., 1996; McKay et al., 2016), we set 8000 years ago as the most likely timing of grounding line retreat over RISP and WGZ. We also examine the preceding and following 1000 years to account for uncertainty in that timing. For each model run we varied the value of $A$ in increments of $10^{-7}$ from $10^{-7}$ to $2\times10^{-5}$ and varied the value of $N_o$ in increments of 0.0025 from 0 to 0.6. To test which values of $A$ and $N_o$ fit observations, we compared the $Fm$ and TOC resulting from our calculations of $^{14}C$ and $^{12}C$ to measured values at WGZ and RISP and noted which values of $A$ and $N_o$ produced $Fm$ and TOC values that fell within the maximum and minimum observed values. Values of $A$ that fit the data ranged from $3 \times 10^{-7}$ to $1.61 \times 10^{-5}$ g/yr per 100 grams of dry sediment. The upper range of these values is comparable to the accumulation rate of organic carbon in modern Ross Sea sediments, of the order of $10^{-5}$ g/yr per 100 grams of dry sediments (recalculated from data in Demaster et al. [1992] and Frignani et al. [1998]).

With parameter ranges for $a$ and $A$ constrained using sample radiocarbon data from RISP, WGZ, and modern amphipods, we then ran the model for all subglacial cores for which radiocarbon has been measured. We varied $A$ over the range determined from RISP and WGZ, varied $N_o$ from 0.08 g to 0.5675 g (normalized to a 100 g sample of dry sediment), and varied the length of exposure to sub-ice shelf conditions as well as the time period of subsequent subglacial conditions from 0 to 8000 years each. As with the model runs for RISP and WGZ, we checked whether each model run was compatible with our measurements by comparing the calculated $Fm$ to the measured $Fm$. Although the samples were collected at the same field sites in the same years, $Fm$ and TOC could not be measured from the same sample; therefore, we tested whether modelled TOC fell within the maximum and minimum measurements of TOC for a given sampling location. Because we only had one measurement of TOC from BIS, we combined TOC measurements from BIS and KIS, and used maximum and minimum values of TOC as the bounds on TOC for both BIS and KIS. In our judgement, this approach is justified because values of $Fm$ for both KIS and BIS are similar (Fig. 2a).

# 3 Results

## 3.1 Ice Temperature Analysis

We used thermal modelling in ice to constrain the time since grounding line readvance ($T_i$) over the sites with steep observed basal temperature gradients (Fig. 5). We ran the model for $0 \leq T_i \leq 8000$ years, but because all basal temperature gradients $\geq$ 50 ºC/km – the definition of cold-based ice used in Engelhardt (2004) – occur when $T_i < 4000$ years, we only examine the model runs where $0 \leq T_i \leq 5000$ years. For each site, we compared the modelled basal temperature gradients to the observations and noted which model results fell within 10% of the observed basal temperature gradients. These comparisons suggest that the grounding line advanced over KIS between 2700 and 300 years ago, over BIS between 1400 and 700 years ago, and over UC between 3600 and 100 years ago. Combining the results for timing of grounding line readvance ($T_i$) for all ice thicknesses allows for evaluation of the frequency at which the model predicted a certain value of $T_i$. Using the median as the optimal timing of grounding line readvance ($T_i$) and the 32nd and 68th percentiles as the bounds on the error, the temperature model thus suggests that the grounding line likely readvanced over KIS $1000^{+200}_{-300}$ years ago, over BIS $800 \pm 100$ years ago, and over UC $1500^{+500}_{-200}$ years ago (Fig. 5). UC has the largest spread of possible values of $T_i$, due to one observed basal temperature gradient differing noticeably from the other two. This may be due to the separation of UC from the others by a paleo-shear margin (named Fishhook) (Clarke et al., 2000). We account for vertical advection, but not horizontal advection in the temperature model. Because the Bindschadler Ice Stream is not currently stagnant and the Kamb Ice Stream has only been stagnant for the past ca. 150 years (Retzlaff and Bentley, 1993), temperature modelling cannot be used to determine a precise time since grounding, but rather it provides a more general idea of how long ago grounding occurred. Importantly, the only model runs that produced basal temperature gradients comparable to those measured at KIS, BIS, and UC were those that assumed ice grounding within the last 4000 years. The temperature modelling also allows us to estimate basal ice thickness growth after grounding. The maximum thickness of basal ice for the three locations examined was 15.8 m for KIS, 10.2 m for BIS, and 19.3 m for UC. These thicknesses only account for the accretion of pure ice, and do not include the contribution of any incorporated debris to the total thickness of debris-laden ice. Given the simplicity of our model, these values are reasonably close to the thickness of debris-laden basal ice (ca. 10-20 m) observed in boreholes in this region (e.g., Christoffersen et al., 2010; Vogel et al., 2005).

## 3.2 Ionic Diffusion Modelling

Ionic diffusion modelling of SLW allowed us to constrain $T_i$ better than $T_o$. For each chemical parameter examined, the modelled diffusion profiles that fit the measured concentration profiles were in agreement with regards to exposure time to ocean ($T_o$) and subglacial ($T_i$) conditions (Fig. 6). We were able to fit diffusion profiles for every value of $T_o$ tested, which impeded our ability to eliminate some lengths of $T_o$ and therefore identify the length of time SLW was exposed to the ocean. Conversely, we were successful in constraining the time since grounding line readvance ($T_i$) as the majority of the diffusion profiles that fit the measured porewater concentrations fall within the past 2000 years.

## 3.3 Radiocarbon Modelling

The *Fm* values reported in Kingslake et al. (2018) and used in this study spanned from $0.0143 \pm 0.0004$ to $0.1058 \pm 0.0013$ (Fig. 2a). Ocean cavity samples recovered from RISP and WGZ showed greater spread in values of *Fm* than those recovered from sites below grounded ice. Samples with *Fm* values closest to the modern reference $^{14}C/^{12}C$ ratio (i.e. closest to *Fm* = 1) were recovered from the sub-ice shelf cavity of the Ross Ice Shelf (RISP and WGZ), and samples furthest from the modern reference $^{14}C/^{12}C$ ratio were recovered from below the Kamb and Bindschadler Ice Streams. Even samples taken from the ocean cavity (i.e. WGZ and RISP) contained only 10% or less of radiocarbon compared to the modern standard. The carbon-to-nitrogen ratio (C:N) of the organic matter from the RISP samples was also the closest to typical 6.7:1 ratios measured in the ocean (Redfield, 1958) (Fig. 2c; Fig. 3), suggesting a significant input of marine organic matter consistent with the exposure of this site to the seawater below the Ross Ice Shelf during the Holocene. Whereas the subglacial sediment samples have higher C:N ratios, from 15.4 – 49.4 (Table S1) (Fig. 2c), which is consistent with their organic matter being a mixture of marine organic inputs and a pre-glacial, recalcitrant radiocarbon-dead component which originated from terrestrial C3 plants (Fig. 3). The C:N ratios of grounding zone deposits sampled at WGZ cluster between the subglacial and RISP samples (Fig. 3). Excluding the C:N values from UC, Fig. 2c indicates that the C:N ratios increase with distance from the modern grounding line. The two UC sediment samples are considered outliers because they came from debris-laden ice (Vogel, 2004, p. 61) rather than from subglacial till. Hence, they retained low, marine-like C:N ratios because the process of basal freeze-on incorporated sub-ice shelf sediments right after ice shelf re-grounding.

The *Fm* values for samples collected from sites currently in the ocean cavity differ only slightly from those currently located below grounded ice. For instance, the mean *Fm* of the seven SLW and WIS samples is $0.050 \pm 0.006$ (standard error of the mean), while the corresponding mean and standard error for the six RISP samples is $0.060 \pm 0.011$. If the RISP samples were covered by ice today, it would take only ~1500 years for their average *Fm* to drop to the level of the SLW/WIS samples through radioactive decay alone, which is remarkably similar to our estimates for how long ago the grounding line readvanced over SLW and WIS. The difference in *Fm* among all subglacial (SLW, WIS, KIS, and BIS) and sub-ice shelf samples (WGZ, RISP) is statistically insignificant based on the linear mixed-effects model (p-value of 0.141, intraclass correlation coefficient of 0.565). The two groups only become statistically distinguishable from each other if we allow radioactive decay to occur in the samples below grounded ice for a period of at least an additional 1200 years. *Fm* values at KIS and BIS are similar to each other but differ from those at WIS and SLW (Fig. 2a), which are also similar to each other. The linear mixed-effects model indicates that KIS/BIS values are statistically distinguishable from the WIS/SLW *Fm* values (p-value of $2.11 \times 10^{-5}$). Overall, the sample-to-sample variations in *Fm* are relatively large compared to any variability due to differences in geographic settings of this sample population.

We used radiocarbon modelling to estimate the duration of ocean exposure following grounding line retreat ($T_o$) and the time since ice shelf re-grounding ($T_i$) (Fig. 7). Unfortunately, the radiocarbon modelling results are not very sensitive to

$T_i$ because the main process changing the simulated $Fm$ of sediments after grounding is radiocarbon decay. That decay has a half-life of ca. 5000 years, or about half of the entire duration of the Holocene and much longer than values estimated for $T_i$ through temperature or ionic diffusion modelling (0-2000 years for SLW, 100-3600 years for UC, 300-2700 years for KIS, and 500-1400 years for BIS). Thus, we use the results from temperature and ionic diffusion modelling to constrain $T_i$.

Additionally, we used the 32nd to 68th percentiles of the results from the temperature and ionic diffusion model results to further constrain the results of $T_o$ found through radiocarbon modelling by only considering radiocarbon model matches where $T_i$ falls within the range determined for each respective area. We then calculated the time of grounding line retreat by adding together $T_o$ and $T_i$ for each radiocarbon model run that produced a model match (Fig. 8). For WIS, where we could not perform ionic or temperature modelling to constrain $T_i$, we used results from the temperature modelling at UC, which is located only a few kilometers away, but across the current shear margin of Whillans Ice Stream. We combined the $T_i$-constrained radiocarbon model results from every core at each site to estimate the peak of radiocarbon model matches for the timing of grounding line retreat at that site. The peak of radiocarbon model matches for timing of grounding line retreat over KIS and BIS were similar: $1800^{+2700}_{-700}$ years ago for KIS and $1700^{+2800}_{-600}$ years ago for BIS (Fig. 8; Fig. 9). The radiocarbon model matches for timing of grounding line retreat over SLW and WIS were slightly more distributed, with peaks occurring at $4300^{+1500}_{-2500}$ and $4700^{+1500}_{-2300}$ years ago, respectively (Fig. 8; Fig. 9).

Radiocarbon modelling of the two cores from SLW produced slightly different results (Fig. 7a). These differences may be attributed to the difference in coring methods employed and by the apparent heterogeneity in sediment $Fm$. The first core (SLW-PEC-1-34-35cm) was collected using a percussion corer, whereas the second (SLW-1 MC1B 0-8 bulk) was collected using a multicorer. The multicorer was designed to preserve and collect the surface sediments, whereas the percussion corer (which was acting as a gravity corer due to data communication issues) probably entered the sediment with backpressure in the barrel, thus blowing away soft surface sediments. Due to inefficient vertical mixing of sediments and the lack of evidence for erosion or deposition at this site (Hodson et al., 2016), we expect that these surface sediments were deposited when SLW was in a marine environment. Thus, the differences in $Fm$ may result from the surface sediments being present in one core, but not the other.

## 4 Discussion

### 4.1 Post-LGM Grounding Line Position

To cast our results in a regional context, we created a schematic diagram of grounding line positions in the Ross Embayment for the past 20,000 years (Fig. 9a). The grounding line along the flow line of the Bindschadler Ice Stream began retreating before 14,700 years ago and remained on the outer continental shelf until at least 11,500 years ago (Bart et al., 2018). The grounding line was then located at Roosevelt Island 3,200 years ago (Conway et al., 1999) before retreating beyond KIS and BIS $1800^{+2700}_{-700}$ and $1700^{+2800}_{-600}$ years ago respectively; it then readvanced over KIS $1000^{+200}_{-300}$ years ago and over BIS $800 \pm 100$ years ago. The grounding line along the Transantarctic Mountain side of the Ross Embayment began

retreating from its LGM position south of Coulman Island ca. 13,000 years ago (Anderson et al., 2014). This retreat was on average fairly rapid, as evidenced by the fact that the grounding line reached WIS $4700^{+1500}_{-2300}$ years ago. Our estimates for grounding line retreat along the flowline of the Whillans Ice Stream are in agreement with the estimates of grounding line retreat over WGZ (Venturelli et al., 2020). Venturelli et al. (2020) estimate that the grounding line retreated over WGZ 7500 – 4800 years B.P, which is only slightly earlier than our estimates of grounding line retreat over SLW ($4300^{+1500}_{-2500}$ years ago) and WIS ($4700^{+1500}_{-2300}$ years ago). This timing is consistent with the fact that SLW and WIS are roughly 100 and 300 km upstream of WGZ. The schematic for grounding line position along the Whillans Ice Stream flow line does not agree with all age constraints found in the Transantarctic Mountain Region (Fig. 9a). This could be because glaciers in the Transantarctic Mountains (for which the exposure ages were measured; Spector et al. [2017]) have a delayed response to grounding line retreat in the Ross Embayment, or because the grounding line retreated faster in the central Ross Embayment than along the sides. Grounding line readvance also occurred relatively swiftly. Timing of this readvance ($1500^{+500}_{-200}$ years ago for WIS and $1100^{+200}_{-100}$ years ago for SLW) is coincident with the grounding of Crary Ice Rise 1100 years ago (Bindschadler et al., 1990). Although Crary Ice Rise is significantly seaward of SLW, it is situated on a pronounced bathymetric high. Therefore, it is plausible that by grounding first, it provided backstress (Still et al., 2019), allowing ice thickening and slow-down to aid the process of grounding line readvance for the Whillans Ice Stream (Fried et al., 2014).

**4.2 Ancillary Evidence Supporting Recent Grounding Line Readvance**

Our analyses suggest that the modern configuration of grounding line positions in the study region has been attained relatively recently. This inference is consistent with the conspicuous absence of grounding zone wedges (GZWs) revealed by detailed seismic surveys at the mouth of the Kamb and Whillans Ice Streams (Horgan et al., 2013b, 2017). These asymmetric sedimentary ridges can form quite rapidly during grounding zone stillstands (Simkins et al., 2018). For instance, the height of the massive Whales Deep GZW in the eastern Ross Sea grew by about 0.1 m per year in the last ca. 1000 years of its formation after growing nearly an order of magnitude slower over the prior ca. 2000 years (Bart and Tulaczyk, 2020). Assuming this range of GZW growth rates of 0.01-0.1 m/yr, in one millennium of GZW stillstand, GZWs can achieve heights of 10-100m. GZWs of such height would be detectable with the active-source seismic methods employed by Horgan et al. (2013, 2017). Hence, the lack of seismic evidence for GZWs at the grounding zones of Kamb and Whillans Ice Streams corroborates the inference that the modern grounding line positions of these ice streams have not been attained until very recently.

The idea that the lower part of the Whillans Ice Stream grounded only recently is also consistent with attributes of the microbial ecosystem discovered in Whillans Subglacial Lake (Christner et al., 2014). Ammonium is the predominant dissolved inorganic nitrogen compound in the lake water column, which also hosts a high abundance of nitrifying microorganisms that obtain energy for chemosynthetic growth through oxidation of ammonia and nitrite (Christner et al., 2014). The source of ammonium for this community is diffusional flux from underlying sediment, facilitated by the activity

of heterotrophic organisms which release ammonium via organic matter decomposition. While the abundant functional groups in the sediments shift to types associated with sulfur oxidation (Purcell et al., 2014) and methane oxidation (Michaud et al., 2016a) with depth, a diversity of heterotrophs exist in both the water and throughout the sediments sampled (Achberger et al., 2016). Similar phylotypes were also detected in sediments from KIS (Lanoil et al., 2009). Glacial meltwater contains no significant quantities of ammonia, and glacial erosion and grinding of minerals is not a significant source of nitrogen compounds (Tranter, 2014). Thus, a nitrifying microbial ecosystem in a subglacial lake, particularly one that is known to experience flushing of dissolved solutes (including nitrogen) from its lake waters every several years (Tulaczyk et al., 2014; Vick-Majors et al., 2020), requires a significant source of bioavailable nitrogen. A recent advance of the ice sheet over sub-ice-shelf sediments like the ones sampled at RISP offers an attractive explanation for the subglacial source of nitrogen fueling the microbial ecosystem found in Whillans Subglacial Lake (e.g., Fig. 3). It is well established that decomposition in organic-poor marine sediments can yield extremely low C:N ratios due to retention of ammonia on clay particles accompanied by the escape of carbon dioxide formed during oxidation of organic carbon stored in sediments (e.g., Müller, 1977). Recent analyses of fluorophore components identified in fluorescent fractions of dissolved organic matter in SLW sediments, while not conclusive, support this notion, indicating characteristics of humic mixtures for coastal environments and marine sediments as well as Antarctic mountain glaciers and lakes (Vick-Majors et al., 2020). The ammonium-dominated, nitrifying microbial ecosystem of Whillans Subglacial Lake may, thus, be living off the legacy of marine organic matter stored in subglacial sediments for a relatively short period of time since the grounding line readvanced over this region.

The mechanism described above also provides an explanation for the seemingly puzzling fact that the sub-ice shelf (RISP) sediment samples, which are exposed to seawater even now, have low C:N ratios which are characteristic of marine sediments (Müller, 1977), but have a very low $Fm$ (0.06 on average). The RISP signature may be caused by the fact that much of the carbon, including radiocarbon, associated with young, recently produced organic matter, was part of labile organic molecules and hence, was preferentially digested during decomposition and released as carbon dioxide. This process, coupled with the fact that ammonia and ammonium are produced commonly in marine sediments through decomposition of marine organic matter rained out from the photic zone, can lower the C:N ratio of the sediments while removing some fraction of radiocarbon and leaving behind radiocarbon associated with more recalcitrant organic compounds which are radiocarbon-dead. The position of most subglacial sediment samples on the $\delta^{13}C$ – C:N plot (Fig. 3) is consistent with the bulk of their organic matter originating from pre-glacial terrestrial C3 plants rich in recalcitrant components such as cellulose, lignin, or sporopollenin. The marine input which is responsible for the presence of radiocarbon (Kingslake et al., 2018; Venturelli et al., 2020) would only make up a small proportion of the total organic matter, and thus does not cause the subglacial sediments to display a marine signature in Fig. 3.

This interpretation is further corroborated by the fact that the two sediment samples from UC, which were melted out of basal ice rather than being sampled from beneath ice (Vogel, 2004, p. 61), show C:N ratios almost as low as those observed in the modern sub-ice shelf sediments of RISP and lower than those observed at the modern grounding line

sediments of WGZ which likely receive an influx of subglacial sediment (Fig. 2c; Fig. 3). Microbial activity, including microbial consumption of nitrogen, is either nil or very slow in sediments incorporated into basal ice as compared to subglacial sediments (Montross et al., 2014). Thus, we interpret that UC's basal ice formed through freeze-on after ice shelf re-grounding took place in the Late Holocene, and that the freeze-on process incorporated sediments containing fresh marine organic matter with a low C:N ratio into the ice. Incidentally, this interpretation of observed low C:N ratios in the two

sediment samples melted out from the basal ice of UC inspired our approach to modelling high basal temperature gradients resulting from recent ice shelf re-grounding.

Additional evidence supporting recent grounding line readvance in the Ross Sea sector has been reported in previous studies. Currently the Siple Coast ice streams are thickening (Joughin and Tulaczyk, 2002), which is consistent with ice sheet advance. Furthermore, the Siple Coast ice streams have experienced stagnation and reactivation in the past

thousand years (Catania et al., 2012), which could potentially be part of the ice shelf grounding process. After examining folds within the Ross Ice Shelf, Hulbe and Fahnestock (2007) concluded that the Whillans Ice Stream must have stopped flowing around 850 years ago. However, we provide an alternative explanation for the folds in the ice layers by positing that they did not form as the ice stream slowed down, but rather as a result of ice shelf grounding. Evidence supporting recent floatation of the lower part of Kamb Ice Stream found by Catania et al, (2005, 2006) is consistent with the grounding line

readvancing to its modern position within the last few hundred years. Finally, very recent grounding of a thin ice shelf produces steep basal temperature gradients, which should result in a rapid basal freezing that may be responsible for the observed frozen-on basal layers found in KIS boreholes (Christoffersen et al., 2010; Vogel et al., 2005) and at WGZ (unpublished data).

## 4.3 Patterns of Grounding Line Retreat in Ross Sea Embayment

There have been alternative suggestions on the style of post-LGM grounding line retreat in the central Ross Sea (Ackert, 2008; Bart et al., 2018; Conway et al., 1999; Halberstadt et al., 2016; Kingslake et al., 2018; Lee et al., 2017; Lowry et al., 2019; McKay et al., 2016; Prothro et al., 2020; Spector et al., 2017), due to few reliable age constraints from areas covered by the Ross Ice Shelf and the ice sheet itself (Anderson et al., 2014). Some conjectures followed the "swinging gate" model (Conway et al., 1999) whereby the grounding line along the Marie Byrd Land side of the Ross Embayment

stayed put ("hinged") near the King Edward VII Peninsula while swinging back along the Transantarctic Mountains on the other side of the Ross Embayment. Others have proposed the "saloon door" model (Ackert, 2008), which envisions that the grounding line began retreating first in the central Ross Embayment and the sides caught up later. Our results are broadly consistent with either of these conceptual models. Exposure age dating along the Scott Coast of the Transantarctic Mountains indicates that the grounding line reached Beardmore and Shackleton glaciers ca. 8000 years ago (Spector et al.,

2017). We find that the grounding line retreated over SLW $4300^{+1500}_{-2500}$ years ago, which would suggest that the grounding line retreated faster along the Transantarctic Mountains, as described in the "swinging gate" model. Conversely, the fact that the grounding line retreated over SLW and WIS earlier than KIS and BIS rather than at the same time could suggest that the

grounding line retreated followed a pattern more in line with the "saloon door" model. Although our age constraints from the Siple Coast ice streams provide added information about grounding line positions, they do not provide constraints on the geometry of the grounding line during early stages of retreat.

These two models have been the enduring paradigms of post-LGM grounding line retreat in the Ross Embayment, however, we suggest that they are too simplistic a representation of grounding line activity because they treat the ice sheet which retreated across the Ross Embayment as a single entity. Currently the Ross Ice Shelf is ungrounded and therefore does behave as a single entity. But there is no reason to expect a uniform retreat from a marine ice sheet that is sitting on a bed with variable bathymetry (Fretwell et al., 2013; Tinto et al., 2019). Using the logic of the marine ice sheet instability (Weertman, 1974), we would expect the grounding line to retreat faster in the troughs and to linger on the bathymetric highs. Recent examinations of geomorphic features in front of the current Ross Ice Shelf edge indicate that the post-LGM grounding line retreat initiated in troughs and left behind transient ice rises (Anderson et al., 2019; Halberstadt et al., 2016; Prothro et al., 2020). We propose extending that idea and speculate that the rapid grounding line retreat seen along the Transantarctic Mountains between 8,600 years ago and 8,000 years ago (Spector et al., 2017) was facilitated by a relatively deep trough visible in the bathymetry below the Ross Ice Shelf (Fig. 1) (Fretwell et al., 2013; Tinto et al., 2019). Because this trough leads almost directly to the Whillans Ice Stream, this could explain why the grounding line retreated earlier over SLW and WIS than KIS and BIS. We believe that the mechanism of grounding line retreat is much more sensitive to bathymetry than is represented in the two canonical models of post-LGM grounding line retreat.

## 4.4 Holocene Climate-Driven Grounding Line Fluctuations

There are varied ideas in the scientific literature as to the mechanisms causing the Holocene grounding line advance in the Ross Embayment. Kingslake et al. (2018) proposed that the grounding line readvance was dominantly due to glacioisostatic rebound following unloading from ice sheet thinning and retreat. Lowry et al. (2019) suggest that this retreat is earlier than, and incompatible with, estimates of ice surface lowering in the Transantarctic Mountains as determined by exposure age dating. Here, we propose that both the grounding line retreat and readvance is primarily attributable to Holocene climate variability. The timing of grounding line retreat (readvance) coincides with atmospheric warming (cooling) inferred from the WAIS Divide ice core (Fig. 8) (Cuffey et al., 2016; Cuffey, 2017). Radiocarbon dating of elephant seal skins found on raised beaches along the Victoria Land Coast indicate two periods during the Holocene when ocean temperatures in this region were warm enough to suppress sea ice formation (Hall et al., 2006). The timing of the first warm period (6800 – 4500 years ago) corresponds reasonably well to the grounding line retreat over SLW and WIS, and the second (2300 – 1000 years ago) coincides with grounding line retreat over KIS and BIS. Following the most recent warm period, ocean temperatures cooled and sea ice cover expanded (Hall et al., 2006). The timing of this ocean cooling corresponds to the cooling period inferred from $\delta^{18}O$ measured in the WAIS Divide ice core (Cuffey et al., 2016; Cuffey, 2017) and is consistent with our estimates of grounding line readvance over KIS, BIS, and SLW. This suggests that the

grounding line readvance was caused by ocean cooling. This is in agreement with results from WAIS simulations indicating Holocene grounding line positions to be most sensitive to ocean temperatures (Lowry et al., 2019). Incidentally, although Kingslake et al. (2018) concluded that the grounding line movement was primarily due to glacial isostatic adjustment, sensitivity testing presented in their supplemental materials show that the timing of grounding line retreat and readvance is sensitive to ocean forcings. Our results from temperature diffusion modelling place the most likely timing of grounding line readvance over WIS during the second warm period. However, given the error it is possible that the re-grounding at WIS may have started during the earlier cool period (3800- 2300 years ago).

Further evidence for ocean temperatures driving grounding line motion is that our results are compatible with warming and cooling in the Ross Sea as indicated by analysis of diatoms in sediment cores (Cunningham et al., 1999). Cunningham et al. (1999) found a period of Ross Sea warming from 7000 to 3200 years ago, which coincides with the timing of grounding line retreat over SLW and WIS, and KIS and BIS within error. Cunningham et al. (1999) additionally proposed a period of cooling from 3200 years ago to present, which coincides with grounding line readvance over all four of our field sites. The proposed high sensitivity of WAIS grounding line positions to relatively small Holocene climate variability may have implications for projections of grounding line behavior during the 21[st] Century as the temperature changes that coincided with the extensive Holocene grounding line retreat are on par with projections of temperature changes by the end of this century (Fig. 8) (Collins et al., 2013; Cuffey et al., 2016; Cuffey, 2017).

### 4.5 Revisiting the Unicorn Paradox

A puzzling observation noted by Engelhardt (2004) about observed basal temperature gradients from the Siple Coast was the difference in basal temperature gradients at UC and WIS. UC is completely surrounded by the Whillans Ice Stream, however, the basal temperature gradients measured at UC were much steeper than those measured a few kilometers away at WIS. Engelhardt (2004) reasoned that cold ice at the bottom of UC could not have formed locally, but rather must have flowed there from Kamb Ice Stream during a proposed "super-surge" event. Given that the grounding line retreated beyond this area within the Holocene (Kingslake et al., 2018), we conjecture that the steep basal temperature gradient is instead a transient signal resulting from recent re-grounding of an ice shelf. If this is correct, Engelhardt's super-surge event was simply due to this part of the ice sheet experiencing transient ungrounding and re-grounding as the grounding line first retreated upstream, and then subsequently readvanced over the sites where Engelhardt measured ice temperature profiles. We estimate the grounding of UC to have occurred between ca. 3600 and ca. 100 years ago. This time frame broadly fits within the range of values of $T_i$ found for SLW. The difference in basal temperature gradients between UC and WIS can be explained by the velocity of the ice. UC is frozen to the bed and has very low surface velocities. Contrarily, WIS has surface velocities of several hundred meters per year. Thus, the ice at the base of WIS reflects migration from upstream and does not record the thermal effects of the recent ice shelf re-grounding in the same way that the slow-moving ice column of UC still does.

## 5 Conclusions

In this study, we use several lines of evidence to refine the current understanding of grounding line activity in the
Ross Sea after the LGM. Modelling subglacial radiocarbon concentrations allows us to estimate the length of ocean
exposure ($T_o$) experienced by our field sites following grounding line retreat, and modelling of ice temperature and sediment
porewater chemistry data enables us to assess the timing of grounding line readvance ($T_i$). Kingslake et al. (2018), who first
proposed that the grounding line in the Siple Coast region retreated past its modern positions after the LGM, favored the
explanation that the retreat resulted from a dynamic overshoot and readvance was driven by glacioisostatic rebound.

Here, we propose an alternative model whereby the grounding line retreated over our field sites as late as the mid-
to late-Holocene and subsequently readvanced during the late Holocene in response to climate cooling during the last 1000 –
2000 years (or 3800 – 2300 years ago in the case of WIS). Grounding line advance during the late Holocene occurred in
spite of the fact that the WAIS Divide ice core shows a ~20% drop in ice accumulation rate over the past 2000 years, from a
maximum reached around 4000 years B.P. (Buizert et al., 2015) – that is when our data suggest grounding line retreat. This
counterintuitive relationship between ice input rates and grounding line motion places emphasis on ice-ocean interactions as
the process capable of translating modest Holocene climate changes (corresponding to temperature variations of less than
2ºC at WAIS Divide) to grounding line migration of hundreds of kilometers (Lowry et al., 2019). By suggesting strong
climatic sensitivity with regard to both retreat and advance, our hypothesis raises further concern for accelerated future
grounding line retreat with increasingly warmer sub-ice shelf oceanic input. It is now recognized that at least some sections
of the Greenland ice sheet retreated during the mid-Holocene climate optimum, and readvanced during the late Holocene
cooling (Vasskog et al., 2015). Further investigations into the relationship of these Antarctic and Greenland ice sheet
fluctuations to Holocene climate variability present an opportunity to reveal the sensitivity of these ice sheets to the slightly
warmer climate states that may be reached in the very near future.

Our results are conjectural, largely because they are based on samples and measurements collected for other
research reasons. However, our study highlights the value of maintaining archival materials because we were able to glean a
greater understanding of grounding line movement in the Ross Sea by applying new modelling approaches to previously
published data collected in different locations from multiple drilling projects over a period of more than 40 years. Future
focused studies may be able to test our hypothesis. Similar efforts should also be aimed at other Antarctic ice sheet margins
where it may be generally assumed that the grounding line was insensitive to Holocene climate variability, simply because
no positive evidence has yet been collected. Insights into the response of Antarctic grounding lines to Holocene climate
changes will inform projections of Antarctic ice sheet evolution under near-future climates, regardless of whether such
insights will indicate high climate sensitivity or a general lack of climate sensitivity.

## Data Availability

With the exception of the TOC, C:N and $\delta^{13}$C measurements which are presented in the Supplemental Material, all data analyzed in this study has been previously published.

## Author Contribution

SN co-designed this research, performed analysis, and wrote the manuscript. ST co-designed this research, performed analysis, and contributed to the writing and editing of the manuscript. NS performed radiocarbon analysis. RS, RP, JC, and ST collected SLW and WGZ sediment samples and performed sedimentological analyses. ST, RP, JM, and RS wrote proposals that funded the collection of sediment samples from SLW and WGZ. NS, JC, RS, JM, and RP contributed to the writing and editing of the manuscript.

## Competing Interests

The authors declare that they have no conflict of interest.

## Acknowledgements

This research was supported by NSF grants: 0838947, 0839142, 1739027. Collection of SLW and WGZ sediment cores was funded by the U.S. National Science Foundation as part of the interdisciplinary WISSARD (Whillans Ice Stream Subglacial Access Research Drilling) project. The U.S. National Science Foundation also supported past drilling projects that enabled the collection of RISP, WIS, KIS, and BIS samples. The drilling team from University of Nebraska–Lincoln, the WISSARD traverse personnel, the U.S. Antarctic Program, the New York Air National Guard, and Kenn Borek Air provided technical and logistical support. Dyke Andreasen and Colin Carney at the University of California Santa Cruz stable isotope lab provided measurements of carbon and nitrogen. We are thankful to Ryan Venturelli and Brad Rosenheim for helpful discussions of radiocarbon analysis in Antarctic subglacial sediments. Statistical analysis was aided by John Neuhaus.

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

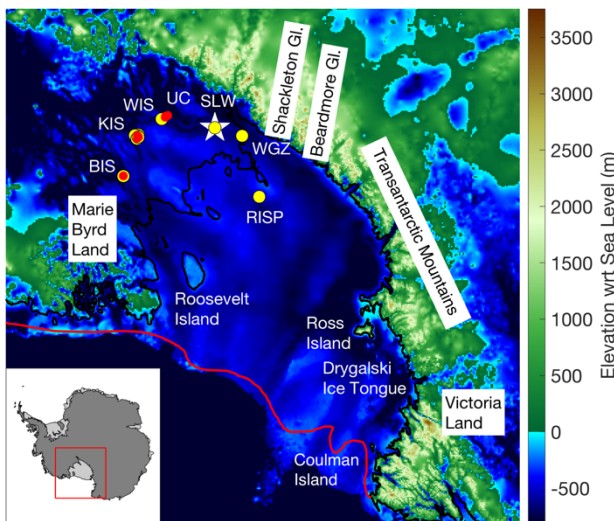

**Figure 1: Location of sites and place names used in this study. Sites where measurements were taken and examined in this study are as follows: RISP (Ross Ice Shelf Project), WGZ (Whillans Grounding Zone), SLW (Whillans Subglacial Lake), UC (Unicorn), WIS (Whillans Ice Stream), KIS (Kamb Ice Stream), and BIS (Bindschadler Ice Stream). Background image uses bed elevation**
**data (Fretwell et al., 2013). Yellow dots denote the location of sediment cores taken for radiocarbon and organic matter analyses. Red dots denote the location of deep ice-temperature profiles examined in this study. White star indicates the location where sediment porewater was collected and analyzed.**

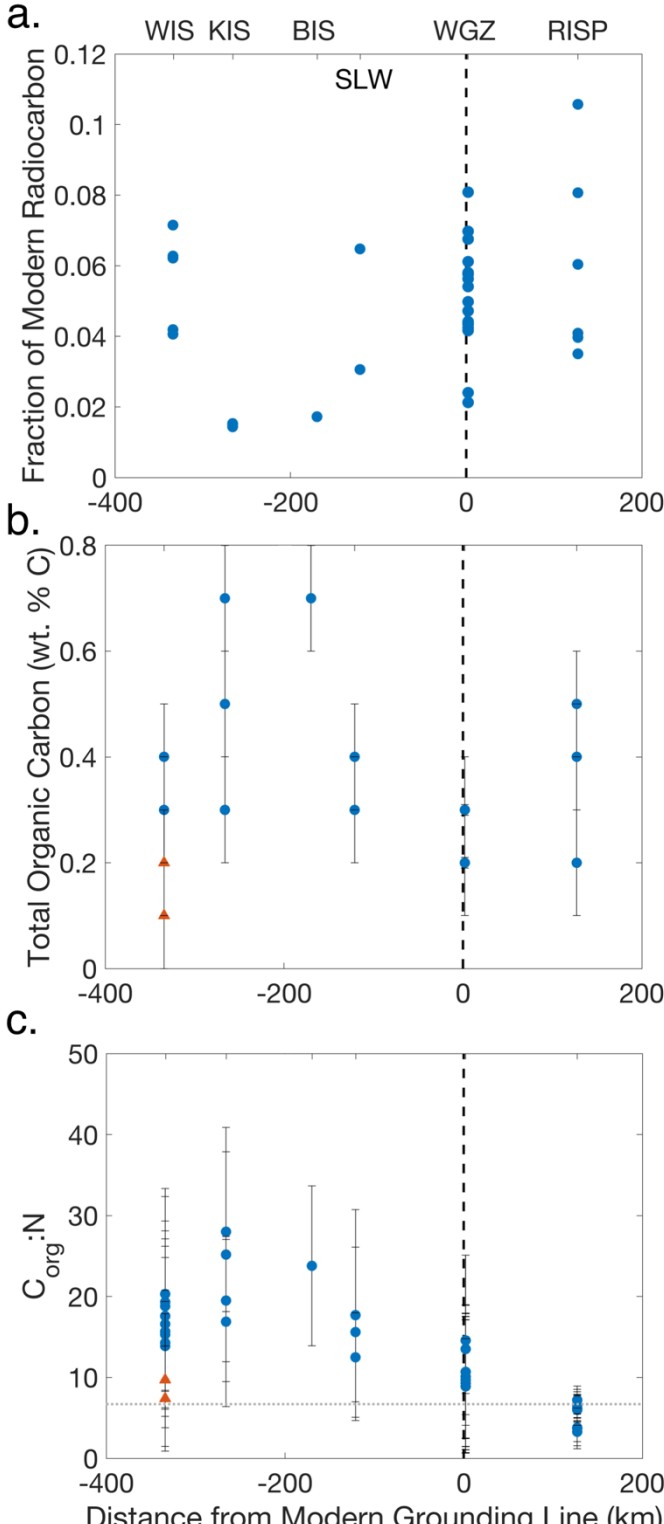

Figure 2: Results of radiocarbon and organic matter analyses for sediment samples taken from beneath grounded ice (SLW, WIS, KIS, BIS), or beneath floating ice (RISP, WGZ), or entrained in basal ice (UC) plotted against sample position with respect to the modern grounding line. (a) Fraction of modern radiocarbon (*Fm*) measured in acid-insoluble organic matter from bulk sediments. Error bars are smaller than the symbols. (b) Total organic carbon (TOC). (c) $C_{org}$:$N_{tot}$ (atom:atom). The dotted gray line represents the typical C:N of the ocean (Redfield, 1958). Orange triangles represent the values for sediments recovered from UC basal ice.

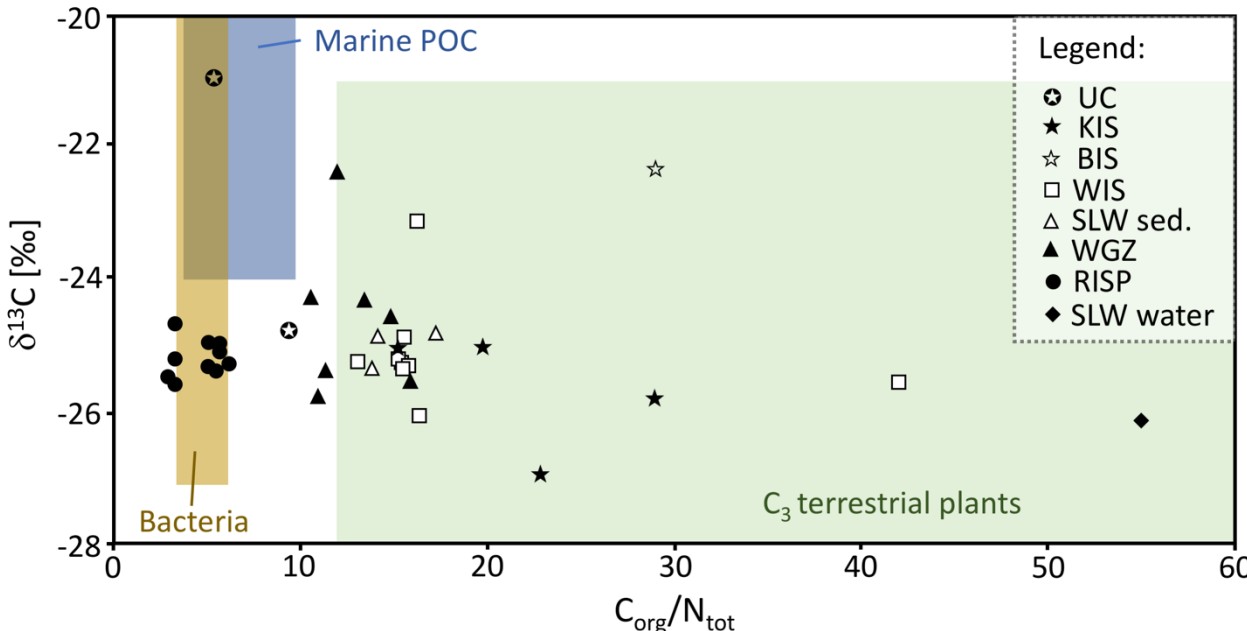

Figure 3: $\delta^{13}C$ plotted against $C_{org}$:$N_{tot}$ (atom:atom). Shaded areas taken from Lamb et al. (2006). Although the RISP and WGZ samples are from a sub-ice shelf cavity, RISP is located mid-ice shelf whereas WGZ is located within the grounding zone. The UC samples are sediments melted out from basal ice, as opposed to collected below ice or an ocean cavity.

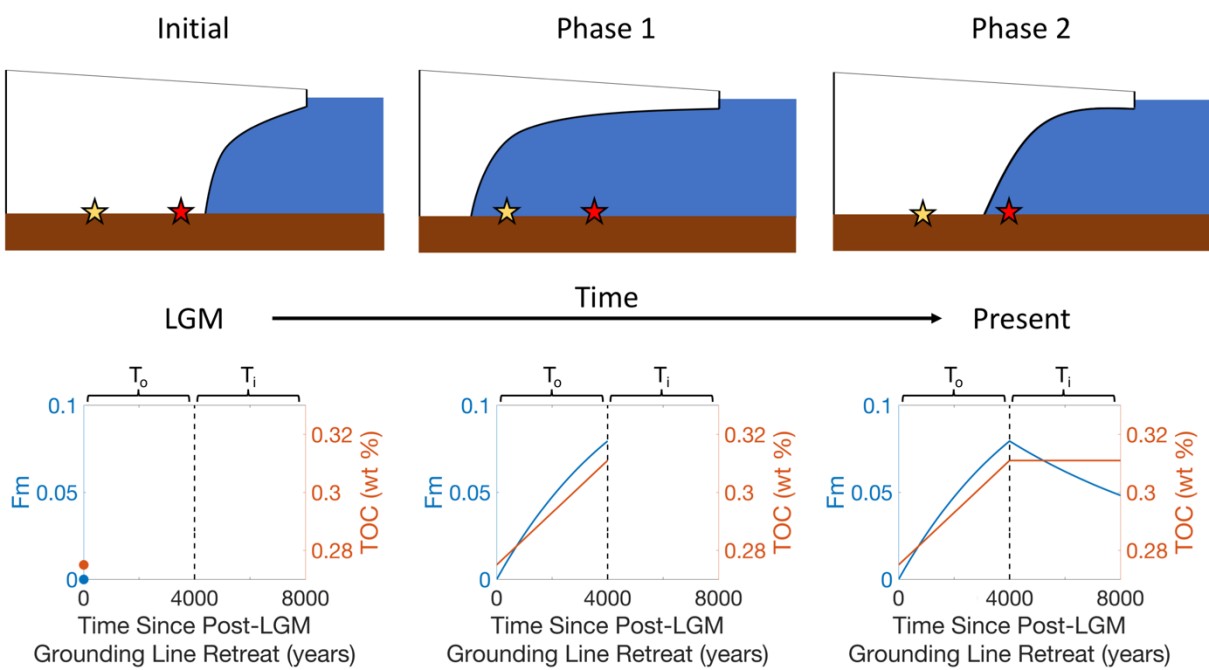

**Figure 4: Schematic of the two-phase model of radiocarbon evolution since onset of LGM grounding line retreat over the field sites. The yellow star represents subglacial field sites, whereas the red star represents sub-ice shelf sites. The evolutions of *Fm* and TOC are shown for subglacial field sites. For this run, both $T_o$ and $T_i$ equal 4000 years; the horizontal axis represents time since post-LGM grounding line retreat past a subglacial site (yellow star). Ocean exposure (when the grounding line had retreated beyond a site) begins at $t = 0$ and is assumed here to last until $t = 4000$ years. Subsequent grounding line readvance occurs at $t = 4000$ years and lasts until the end of the model run ($t = 8000$ years). Accumulation rates for [14]C and [12]C were assumed to be 9.23 x $10^{-18}$ g/yr and 9.0 x $10^{-6}$ g/yr per 100 grams of dry sediments, respectively. Note that the right-hand side axis (TOC) does not start at zero.**

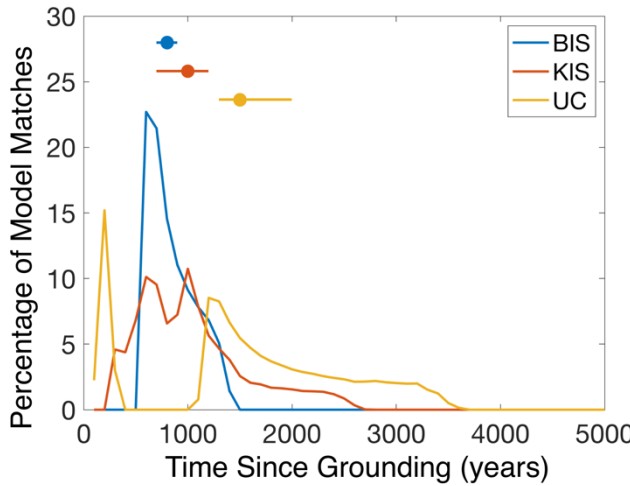

**Figure 5: Results of temperature diffusion modelling.** The y-axis represents the frequency at which the modelled basal temperature gradients fit within 10% of the observed basal temperature gradients (Engelhardt, 2004). The number of observed basal temperature gradients varied between the sites: one observation for BIS, four observations for KIS, and three observations for UC. The dots and lines in the upper portion of the figure correspond to the median and the 32nd to 68th percentiles of the distributions shown in the lower portion of the figure. The total number of model runs performed was 808,000, corresponding to a time window of 8000 years ago to present.

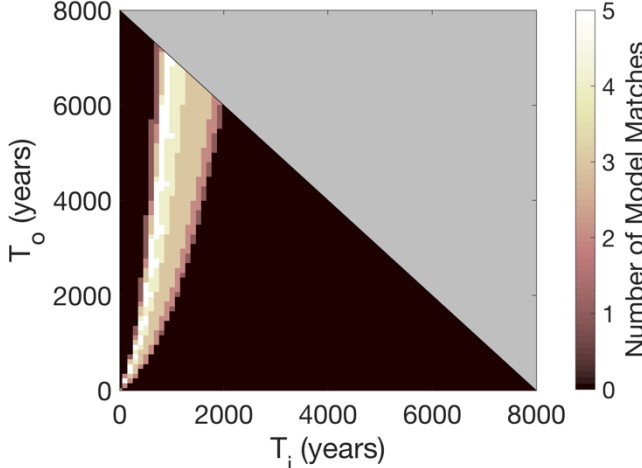

**Figure 6: Stacked results from ionic diffusion modelling of all six chemical parameters.** $T_o$ represents the length of time over which the topmost sediment was exposed to ocean water after initial grounding line retreat and before grounding line readvance. $T_i$ represents the length of time over which the topmost sediment was exposed to subglacial conditions between the grounding line readvance over SLW and now. The number of model runs included in this figure is 19,926.

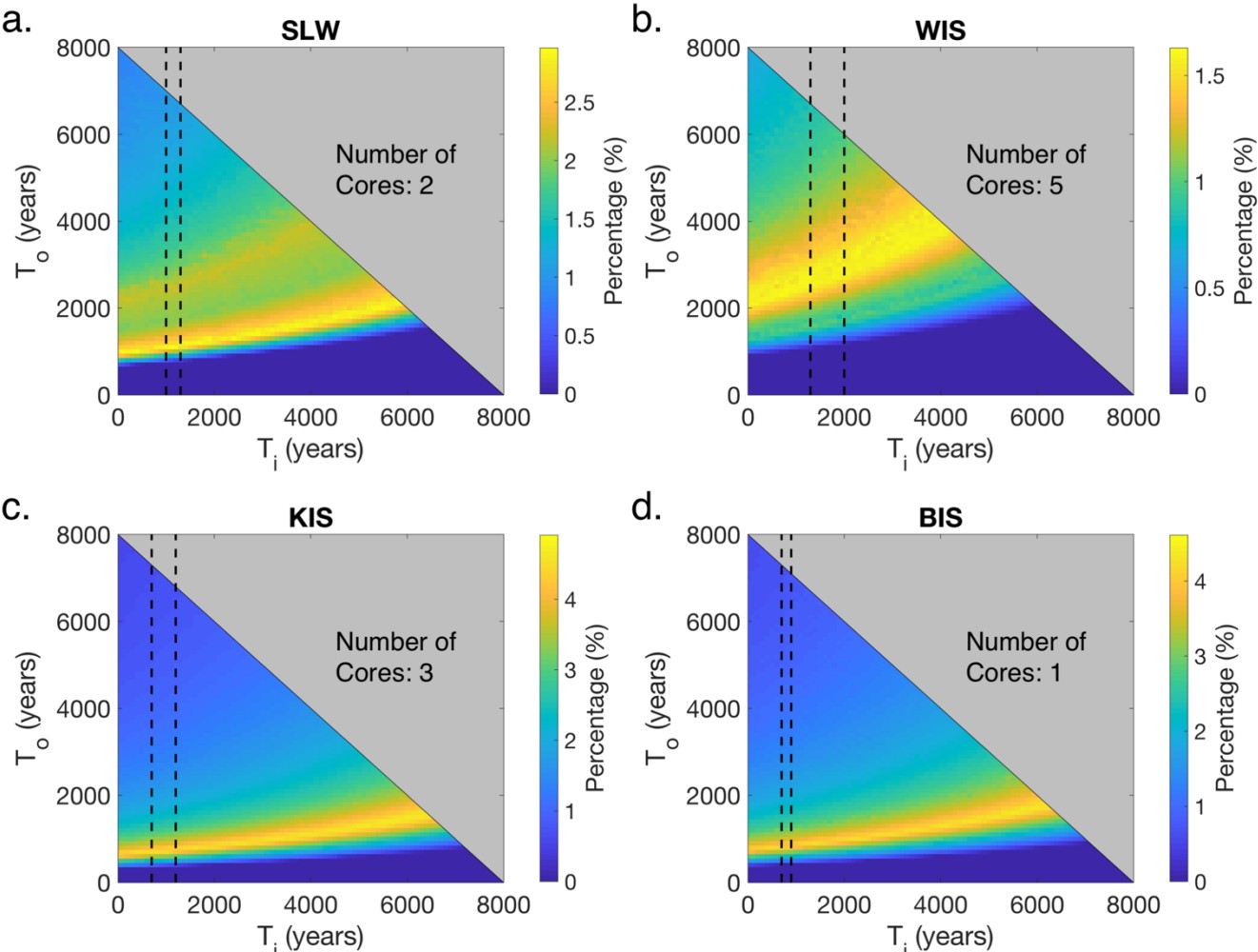

Figure 7: Results of radiocarbon modelling for all eleven subglacial cores. The axes correspond to the length of time the grounding line position had either retreated behind the sites ($T_o$) or advanced over them ($T_i$). The colorbar indicates the percentage of model runs which produced *Fm* and TOC that fell within the observed ranges for each core. A total of 103,495,644 model runs were performed for each core. Results from cores at the same field site are stacked on top of each other. Thus, the total number of model runs for each site is 206,991,288 for SLW, 517,478,220 for WIS, 310,486,932 for KIS, and 103,495,644 for BIS. The large number of model runs (which serve as the denominator in calculations of percentage of model matches) are responsible for the relatively small percentages shown in a-d. (a) SLW (b) WIS (c) KIS (d) BIS. a-d indicate that the model provides better constraint to $T_o$ than $T_i$. For example, (c) shows that at KIS, the model prefers ocean exposure durations ($T_o$) of 600-1800 years but does not constrain the duration of grounding line readvance ($T_i$). The dashed lines indicate the 32[nd] and 68[th] percentiles from the results of the duration of $T_i$ as determined by ionic and temperature diffusion modelling.

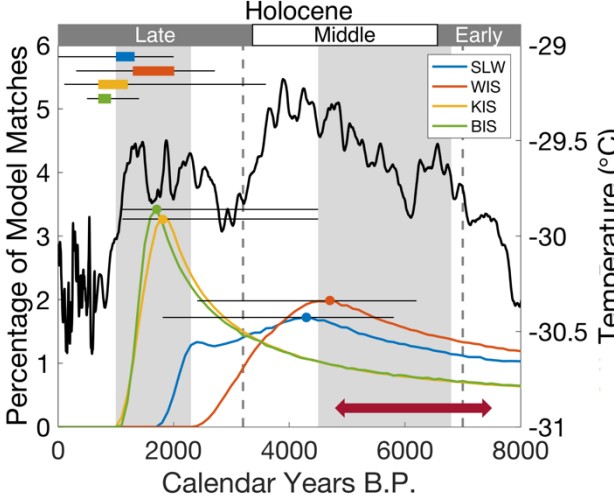

**Figure 8: The estimated timing of grounding line retreat and readvance compared to selected climate data for the study region.** The colored lines (which correspond to the left y-axis) represent the probability distribution of the timing of grounding line retreat ($T_o + T_i$) for each subglacial field site. The lines are calculated by summing $T_o$ and $T_i$ from the model matches in between the dashed lines of the radiocarbon model results shown in Fig. 7. The dots on the peaks of the distributions indicate the optimal timing of grounding line retreat, and the thin black lines indicate the estimate on error. The box and whisker plots in the upper left corner indicate the timing of grounding line readvance over our field sites ($T_i$) estimated from temperature and ionic diffusion modelling. The thick black line (which corresponds to the right y-axis) represents the Holocene history of surface temperature at WAIS Divide (Cuffey et al., 2016; Cuffey, 2017). The gray vertical shaded regions indicate the warm periods of less extensive sea ice in the Western Ross Sea proposed by Hall et al. (2006), and the gray dashed lines bracket the period of Ross Sea warming identified by Cunningham et al. (1999). The maroon arrow indicates the estimated range of grounding line retreat over WGZ based on the ramped-pyrolysis method (Venturelli et al., 2020).

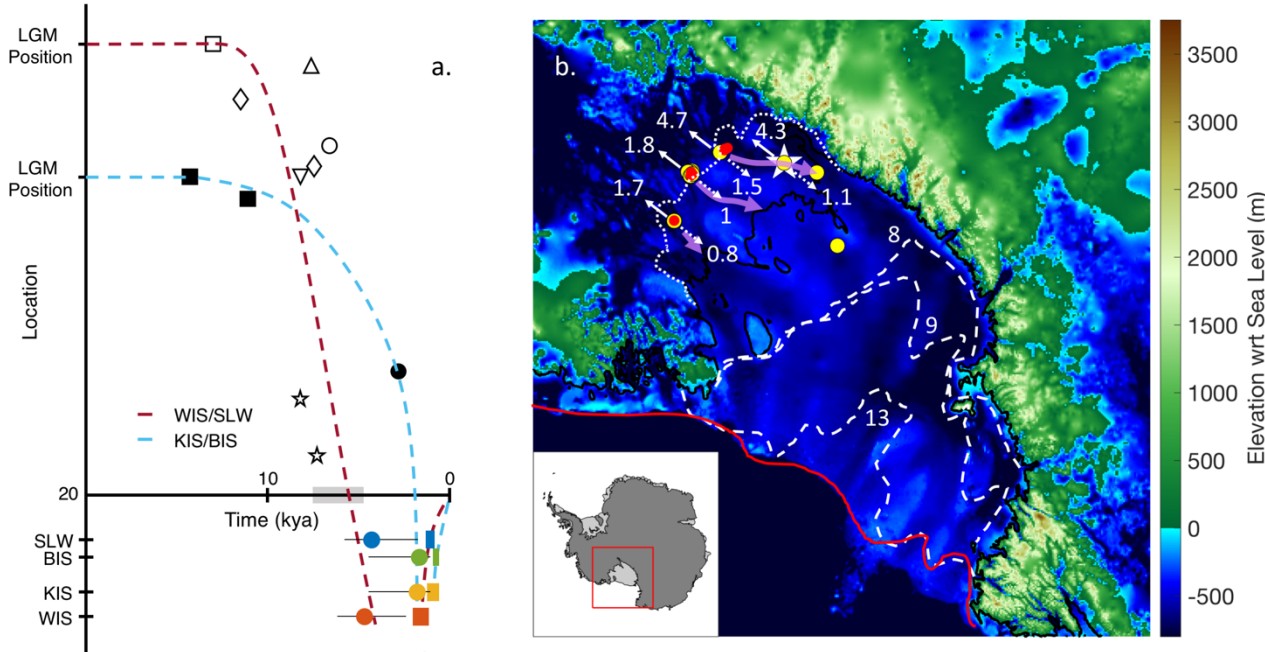

**Figure 9: The evolution of grounding line positions in the Ross Embayment for the past 20,000 years.** (a) Schematic for grounding line position along the flow lines of the Whillans Ice Stream (maroon dashed line) as well as the Bindschadler Ice Stream and Kamb Ice Stream (blue dashed line). The colored dots represent the optimal grounding line retreat timing (error indicated by thin black lines) seen in Fig. 8. The colored bars represent the optimal timing of grounding line readvance (between the 32[nd] and 68[th] percentiles). Symbols represent age constraints on grounding line position from the following studies: Baroni and Hall (2004) (right side up empty triangle), Bart et al. (2018) (solid square), Conway et al. (1999) (solid circle), Cunningham et al. (1999) (empty square), Jones et al. (2015) (empty circle), Licht et al. (1996) (empty diamond), McKay et al. (2016) (upside down empty triangle), Spector et al. (2017) (empty star). Open symbols represent ages along/near the flowline of the Whillans Ice Stream, and solid symbols represent ages along the flowline of the Bindschadler Ice Stream. Gray bar along the grounding line represents grounding line retreat at WGZ calculated in Venturelli et al. (2020). (b) Map-view of Holocene grounding line positions in the Ross Embayment. Background image is bed elevation (Fretwell et al., 2013). Yellow dots denote the location of sediment cores taken for radiocarbon and TOC analysis. Red dots denote the location of deep temperature profiles examined in this study. Cyan diamond indicates the location where sediment porewater was collected and analyzed. Red line indicates LGM grounding line position (Bentley et al., 2014). Dashed white lines indicate grounding line retreat from Lee et al. (2017). The corresponding numbers indicate timing in kya (thousands of years). Dotted white line shows most retreated grounding line position modelled in Kingslake et al. (2018). Solid white arrows indicate timing of grounding line retreat in kya, and dotted white arrows indicate timing of grounding line readvance over those sites. Thick purple arrows indicate flowlines along the Whillans, Kamb, and Bindschadler Ice Streams.

**Table 1: Observed basal temperature gradients reported in (Engelhardt, 2004) and used in the temperature diffusion model.**

| Site | Basal Temperature Gradient (K/km) |
|------|-----------------------------------|
| BIS | -61.2 |
| KIS | –54.6 |
| KIS | –60.5 |
| KIS | –63.4 |
| KIS | –71.3 |
| UC | –83.5 |
| UC | –81.0 |
| UC | –51.5 |

955

**Table 2: Diffusion coefficients and concentrations for chemical parameters examined in the porewater diffusion model. The seawater concentrations are the typical concentrations in standard 35 per-mil seawater, and the meltwater concentrations are from measurements taken in Whillans Subglacial Lake (Michaud et al., 2016a).**

| Chemical Parameter | $D_{sed}$ (m²/yr) | Seawater Concentration *(g/kg) or **(‰) | Meltwater Concentration *(g/kg) or **(‰) |
|--------------------|-------------------|------------------------------------------|-------------------------------------------|
| $Cl^-$ | 0.0113 | *19.353 | *0.125 |
| $SO_4^{2-}$ | 0.00557 | *2.712 | *0.053 |
| $Na^+$ | 0.00699 | *10.76 | *0.121 |
| $Ca^{2+}$ | 0.00416 | *0.412 | *0.017 |
| $\partial^{18}O$ - ($H_2O$) | 0.0162 | **0 | **-38 |
| $\partial D$ - ($H_2O$) | 0.0111 | **0 | **-300.9 |

960