# Peer review of "Did Holocene climate changes drive West Antarctic grounding line retreat and readvance?"

_The Cryosphere, 2020_

## Referee Comment (RC1) · Anonymous Referee #1 · 31 Dec 2020

**General comments:**

Neuhaus and colleagues present a new modelling approach to constrain post-last glacial maximum grounding line behavior in the Ross Sea sector of West Antarctica. In this study, the authors explain previously published (Kingslake et al., 2018) radiocarbon data from subglacial and sub-ice-shelf sediment samples using a two-phase model for radiocarbon input and decay to determine the timing of grounding line retreat beyond sampling sites beneath Whillans, Kamb, and Bindschadler ice streams. The timing of re-advance over the same sites was determined using previously published basal temperature gradients (Engelhardt, 2004), porewater chemistry profiles (at Whillans Subglacial Lake; Michaud et al., 2016), and geophysical evidence (at the grounding zone of Whillans and Kamb ice streams; Horgan et al., 2013; 2017). Given that both the style (Swinging gate vs. saloon door vs. marine-based vs. retreat and re-advance; reviewed in Halberstadt et al., 2016) and timing of grounding line retreat in the Ross Sea Embayment (recently reviewed in Prothro et al., 2020) is the subject of active debate in the community, this re-assessment of previously published data has the potential to draw interest from both modern- and paleo-glaciological researchers. However, several points necessitate significant revision before this manuscript is accepted for publication.

**Major comments on scientific content:**
1. *Assumptions of the radiocarbon model:* The main assumption of this study is that any radiocarbon present in subglacial sediment samples in this region comes from the marine environment. This assumption is supported by two previous studies—Kingslake et al. (2018), which is cited in the main text, and Venturelli et al. (2020) which is only cited in the figures. The authors should include further discussion of these two papers in the explanation of their assumptions to make it clear that this point is well-established in previously published literature. The model assumes that radiocarbon was added to these sediments at a constant rate while exposed to the marine waters (line 17-18 of supplement). This assumption should be justified with respect to the proposed mechanism of radiocarbon input to these sediments (228-229 of main text)—was radiocarbon addition set to a constant rate because it is physically realistic for fecal pellets and faunal necromass to be deposited at a constant rate or because it is the simplest way to model input? How much would a variable rate of radiocarbon input change the timing of grounding line retreat? What is the uncertainty that this assumption imposes on the result? In the independent evolution of $^{12}C$ in this model, further explanation of carbon supply should be included. There is reference to radiocarbon free material on continent with reference to a previous study (Tulaczyk et al., 1998); however the authors set carbon input to 0 after the grounding line re-advances. How would the transport of radiocarbon-free material by the overlying ice streams impact model outputs? A sensitivity test of variable vs. constant carbon inputs would demonstrate that this assumption is sound, similar to what was done for the ice temperature model.
2. *Details about radiocarbon model:* Whereas the authors do a good job of laying out equations used in this model and provide a graphical depiction of model outputs (figure 6, and S1), significant details of the modeling methods are missing, preventing me making a careful assessment of this key piece of the manuscript. The caption of figure 6 provides a percentage of simulated Fm values that match measured values, however nowhere in the figure caption or text is it stated how many model runs were performed. As written, the manuscript lacks explanations for why the (maximum) 4% match between modeled and measured Fm shown in figure six is significant. Does this mean the model only reproduced the measured values 4% of the time?
3. *Proposed mechanism of radiocarbon input:* The authors state that radiocarbon in these sediments "likely" comes from fecal pellets and faunal necromass. Have intact fecal

pellets or diagnostically Holocene macro/micro/meiofaunal parts been observed in previous micropaleontological investigations of these sediments (e.g. Harwood et al., 1989; Scherer, 1991; Scherer et al., 1998; Coenen et al., 2019) or subsequent investigation herein? If micropaleontological evidence for Holocene particulate carbon input does not exist, it should be made clear that this mechanism is assumed, and the assumption should be defended in this discussion of this manuscript.

4. *Uncertainty in retreat ages:* In its current form, the manuscript lacks any quantification of the uncertainty in modeled retreat ages. The interpretation presented in this manuscript suggests grounding line retreat 4,000 years before present at Whillans Ice Stream and 2,000 years before present at Kamb and Bindschadler ice streams. Based on the figures 6 and S1, it seems that the presented values may be the mean output of the radiocarbon model, however it is not clear from the methods or results if this is true. At the very least, it would benefit the manuscript to provide quantitative uncertainty of model outputs. I note that a "frequency of successful model runs" is included in figure eight, but there are no values tied to the color bars. It is therefore impossible to assess the weight of these results or contextualize them with previous studies. Given the noted assumptions in both the radiocarbon and temperature models, a full propagation of uncertainty should be presented to support the presented retreat and re-advance timing. Error bars should be added to the presentation of timing in figures 7 and 8 and discussion should be added in text about whether or not the modeled timing in this study falls within error or previous studies from this region (Kingslake et al., 2018; Venturelli et al., 2020). The authors are specifically using their modeled retreat and re-advance timing to designate a climate-related forcing mechanism instead of previously suggested sea level and glacioisostatic forcings, but without any presentation of uncertainty it is unclear how reliable this alternative explanation is.

5. *Interpretation of geochemical data:* It is well-established that the subglacial sediments in this study contain a mixture of past marine and terrestrial microfossils (e.g. Coenen et al., 2019). Though this point is acknowledged in this manuscript (lines 169 and 301), the authors use bulk geochemical analyses (C:N ratios, Fm, $\delta^{13}C$) to make interpretations about the origin of organic material in the samples herein. At present, the manuscript lacks sufficient discussion of how a multi-source mixture would appear in the results of these geochemical analyses. Furthermore, the assertion of a marine source of organic material is not consistent with the shaded boxes in figure 9. It would benefit the explanation of new data generated herein to compare measured $\delta^{13}C$ values to $\delta^{13}C$ values for particulate organic carbon and sedimentary organic carbon in the contemporary Ross Sea (e.g. Villinski et al., 2000). This information would improve the presentation of data in figure 9, and further explanation should also be included in the discussion.

6. *Temperature model:* In the current form, the manuscript lacks quantification for the uncertainty of re-advance timing. The sensitivity testing (Supplement section 2) indicates that the authors considered uncertainty in assumptions of this model and may therefore be more reliable than the retreat timing determined with the radiocarbon model. The constraint on re-advance timing has wide ranging importance from providing a constraint on Holocene ice dynamics to aiding in the interpretation of microbial data—a point that is very well illustrated in paragraphs on lines 372-394. However, the significance of this important result is overshadowed by a lack of error bars.

Uncertainty must be included for both the timing of retreat and re-advance to assess what new knowledge is presented in this manuscript. Without an explanation of the uncertainty in these age ranges, it is impossible to determine if modeled results are significantly different spatially (i.e., the difference between Whillans and Kamb/Bindschadler ice streams herein), temporally

(i.e., Is there any overlap in the modeled timing of retreat and readvance at any site?), or from the many studies surrounding the chronology of grounding line retreat in this region (most notable to the sites included in this study: Spector et al., 2017; Kingslake et al., 2018; Venturelli et al., 2020). A discussion of all of these points must be included to set this study apart from previous work in this region. If the timing presented in this paper is significantly statistically different than previous studies in this region, the inclusion of uncertainty has the potential to make this paper impactful for a wide-ranging scientific audience.

**Line/Technical comments:**
43: Smith et al., 2019 (doi: 10.1038/s41467-019-13496-5) provided a comprehensive review on this topic and should be cited here.

43: You can/should state that this is the enduring paradigm in the Ross Sea Sector, and that this has been recently challenged [Bradley et al., 2015; Kingslake et al., 2018; Venturelli et al., 2020 (you could even add in Greenwood et al., 2018 if you wanted to include the EAIS portion of the Ross Sea Embayment)]. However, "scientific consensus" is too strongly phrased given the large body of work detailing the active debate in this region. As written, this also contradictory to the "disagreements" discussed in section 4 of the supplement. Some comprehensive reviews (e.g. Prothro et al., 2020 and Halberstadt et al., 2016) of this debate are not cited here or throughout the manuscript, but should be to provide better context on the state of knowledge.

63-66: There should be some detail added about why you conjecture re-grounding as the explanation for observed unsteady thermal state. Other processes that may explain this observation should be mentioned in text or even tested with your temperature model. A demonstration of how you ruled out these other processes would add strength to the statement in this sentence.

75-77: Do you plan to include your MATLAB code in the supplement? The reference here and elsewhere in the manuscript make me curious to see it.

87: Explanation for the assumed ice shelf thicknesses (500-1000 m) is needed. If these are based on the modern thickness distribution of Ross Ice Shelf, a citation should be included.

91-93: Explain why you assume basal freeze on to have occurred after re-grounding. Do you have any age constraint on accretion? Justification for this assumption should be added.

95-99: It would benefit this manuscript to add a brief explanation of why you set 10,000 years as the model boundary. I realize that this is tied to the work of Kingslake et al. (2018), but it would improve the clarity of this paper to add a sentence or two to explain this so the reader does not have to search through another paper for this explanation.

98: Should "advanced" be swapped for "retreated" here?

102: Here and throughout, "Subglacial Lake Whillans" should be changed to "Whillans Subglacial Lake" to align with the official place name established in 2018. Further details can be found with the direct link provided below:
https://geonames.usgs.gov/apex/f?p=138:3:::NO::P3_ANTAR_ID,P3_TITLE:19707,Whillans%20Subglacial%20Lake

136, 144: Provide further explanation for the instantaneous change in boundary conditions. Do you assume that there is no transitional phase in which seawater and subglacial water are mixed as the grounding line re-advances, as has been proposed for grounding lines in this region? (e.g. Horgan et al., 2013 doi: 10.1130/G34654.1)

169: "Meaningless" should be changed to "Chronologically meaningless". The presence of radiocarbon in subglacial sediments was used in Kingslake et al (2018) to challenge the enduring paradigm of grounding line retreat in this region. That alone makes those data meaningful, and your work in this manuscript has the potential to add further value.

176: Sediments intended for geochemical analyses of acid insoluble organic material are conventionally decarbonated using hydrochloric acid. Some explanation should be added about why a different method is used here. Important details about the size of samples (mg? g?) decarbonated using this method are noticeably absent.

225: Space between number and unit (100 g)

228-229: I make note of my questions surrounding your radiocarbon input mechanism above (#3), but the statement about advection here raises further question. Are you assuming that particulate carbon is being advected under Ross Ice Shelf from the open marine environment or elsewhere in the sub-ice-shelf environment?

307: Provide explanation when stating that something is "surprising". If the primary source of sedimentation at your sub-ice shelf sites is melting of basal debris from the overlying ice shelf that you assume accreted following grounding line re-advance, one could expect these samples to be geochemically similar.

334-340: This paragraph makes it seem like you are assuming that sediments present at the surface when your samples were collected are the same sediments that were present at the surface when grounding line retreat occurred. If I am interpreting your assumption correctly, provide some context for why significant sediment accretion would not have occurred in the ~2000-4000 years between your modeled retreat and sample collection.

365: Simkins et al., 2018 (doi: 10.5194/tc-12-2707-2018) would be a good citation to strengthen the grounding zone wedge sentence here.

403-405: Should add an explanation of the proportion of inputs needed to result in the values shown in figure 9.

Figure 2: Error bars for geochemical data (Fm, %TOC, C:N) are needed (and should be propagated through all analyses).

Figure 4: Note how many temperature model runs were performed.

Figure 5: Are these results for ionic diffusion modeling efforts of all chemical parameters noted in Table 1? How many model runs were performed?

Figure 6: It is stated in the caption that the model runs are stacked for each core, but it is not indicated how many model runs were performed.

Figure 7: Add uncertainties for timing indicated by colored lines. Are they within error of measured values of Venturelli et al (maroon error)? It would be interesting to supplement this point in your figure with some explanation in the discussion.

Figure 8: Do the color bars for frequency of successful model runs in 8a have any associated number values? Explanation should be included in the figure caption for what designates a successful model run for grounding line retreat. Figure clarity would be improved by adding a key for the shape points and line colors in 8a rather than an explanation in the caption.

Figure 9: The x-axis label says $C_{org}/N_{tot}$, but the caption says $C_{org}:N_{org}$ It seems from the supplement and methods that only Total Nitrogen (TN) was measured. Can you clarify?

Supplement 143-144: The "Saloon Door" comes from Ackert (2008). This citation should be added in addition to the already included citations for later discussions of this model.

**Below are citations included in this review that are not already included in the manuscript:**
Ackert, R. (2008, October). Swinging gate or Saloon doors: Do we need a new model of Ross Sea deglaciation. In *Fifteenth West Antarctic Ice Sheet Meeting, Sterling, Virginia* (Vol. 811).
Coenen, J. J., Scherer, R. P., Baudoin, P., Warny, S., Castañeda, I. S., & Askin, R. (2020). Paleogene marine and terrestrial development of the West Antarctic Rift System. *Geophysical Research Letters*, *47*(3), e2019GL085281.
Halberstadt, A. R. W., Simkins, L. M., Greenwood, S. L., & Anderson, J. B. (2016). Past ice-sheet behaviour: retreat scenarios and changing controls in the Ross Sea, Antarctica. *The Cryosphere*, *10*, 1003-1020.
Horgan, H. J., Alley, R. B., Christianson, K., Jacobel, R. W., Anandakrishnan, S., Muto, A., ... & Siegfried, M. R. (2013). Estuaries beneath ice sheets. *Geology*, *41*(11), 1159-1162.
Prothro, L. O., Majewski, W., Yokoyama, Y., Simkins, L. M., Anderson, J. B., Yamane, M., ... & Ohkouchi, N. (2020). Timing and pathways of East Antarctic Ice Sheet retreat. *Quaternary Science Reviews*, *230*, 106166.
Simkins, L. M., Greenwood, S. L., & Anderson, J. B. (2018). Diagnosing ice sheet grounding line stability from landform morphology. *The Cryosphere*, *12*, 2707-2726.
Smith, J. A., Graham, A. G., Post, A. L., Hillenbrand, C. D., Bart, P. J., & Powell, R. D. (2019). The marine geological imprint of Antarctic ice shelves. *Nature Communications*, *10*(1), 1-16.
Villinski, J. C., Dunbar, R. B., & Mucciarone, D. A. (2000). Carbon 13/Carbon 12 ratios of sedimentary organic matter from the Ross Sea, Antarctica: A record of phytoplankton bloom dynamics. Journal of Geophysical Research: Oceans, 105(C6), 14163-14172.

---

## Short Comment (SC1) · 4 Jan 2021

Thank you for such a detailed and helpful review. Your suggestions will definitely improve our manuscript.

---

## Referee Comment (RC2) · Anonymous Referee #2 · 15 Mar 2021

**Review of Neuhaus et al – Did Holocene climate changes drive West Antarctic grounding line retreat and re-advance?**

Firstly, my apologies that this review is right up against the deadline that I was given – I had some unexpected personal circumstances.

Secondly, for contextualising some of my several comments on figures it is worth noting that I have a mild but common colour deficiency. It might be helpful for the authors to know that as a general rule when small colour symbols (or thin lines) are placed on top of other colours it can be extremely difficult to distinguish the colours and match them to a key. There is now wide debate on this in scicomms literature and some good reviews available to guide authors e.g. https://www.climate-lab-book.ac.uk/2014/end-of-the-rainbow/ – most of R, GMT, MATLAB etc now have colour-vision-friendly palettes available.

This paper aims to address the question of timing of a grounding line retreat and re-advance in the Ross Sea sector. This has been part of a debate for some time on retreat history of the region but this paper particularly follows on from the work of Kingslake et al (2018) which attempted to constrain retreat timing but which yielded dates for retreat that were inconsistent with significant amounts of other observations.

In this paper, the authors take three main approaches to constraining past timing of retreat and readvance: and in each case comparing model simulations to archived (and some new) measurements from subglacial and sub-ice shelf sample sites. First, they use temperature modelling of the basal ice to constrain the timing of grounding line readvance. Secondly they use iconic diffusion modelling of the topmost portion of subglacial sediment to constrain timing of readvance. And thirdly they use modelling of radiocarbon content of subglacial sediment to constrain the period of ocean exposure between grounding line retreat and readvance. The authors then go on to draw some conclusions on the likey forcing factors and conclude that the drivers for retreat and readvance were climatic rather than delayed ice dynamics and glacioisostatic adjustment, as suggested by Kingslake et al.

The paper is based on an interesting idea which has potential to provide insight into the grounding line history, and I very much like the use of archived samples and long-standing measurements. The results will be potentially of interest to a broad community, but they need further explanation and some additional clarity (especially on uncertainty) before the paper should be accepted for publication.

**Broad issues**
**Model details.** The derivation of model equations is mostly well dealt with but some other details such as numbers of runs, uncertainty (see below), and how to interpret some of the output (especially in some very rich figures), are not yet clear enough.
**Model assumptions** – in a number of places it is difficult to follow if assumptions/choices are made for model simplicity or are based on a comprehensive observational dataset. For example, a single porosity value of 0.4 (line 126) seems surprising: marine sediments vary significantly in porosity and I imagine that subglacial sediments do too. Similarly the supplementary gives a single value for geothermal heat flux but without discussing any likely range or sensitivity of the model to this parametrisation. Whilst the choice of parameters may be helpful it is difficult to assess the model results without knowing the uncertainty bounds created by uncertainty in parametrisation. Some sort of sensitivity or uncertainty analysis of the model is needed to try and understand the range in final exposure/readvance durations. For example, the plots in Fig 6, which I read as a form of

probability density plot translate into single thin lines on figure 7 but without uncertainty bounds included.

**Unmodelled processes:** What would be the effect of sediment accretion and/or deformation following grounding line readvance? Given this is the 'type area' for actively deforming till layers it probably needs some qualitative comment on the likely effects of such deformation on the radiocarbon and/or ionic diffusion observations. If it is only simple shear then there would be no vertical movement but till accretion is possible and this would likely lead to addition of deep interior radiocarbon-dead carbon. I think some qualitative discussion on possible post-depositional changes would be important here, as the implicit assumption is that the sediment being sampled was the same sediment exposed as the cavity was closed.

**C:N rationale and treatment:** The explanation of why C:N is being measured and analysed does not become clear until the Results in Section 3.3, when Figure 9 is called (out of sequence) and the different fields for marine and terrestrial become clear. I suggest putting Fig 9 much earlier and calling it from the methods where it will be possible to explain the concepts behind measuring C:N ratios and del-13C. This explanation can also then make clear the reasoning as to why terrestrial plants field matter (I presume this is because the radiocarbon-dead material is assumed to come from long-dead terrestrial plant material in bedrock beneath the WAIS but I couldn't see it made explicit anywhere). Note also comments below on use of weight:weight and atom:atom ratios on the same plot.

**Presentation of (radio)carbon data on sediments**: It would be helpful to include analytical error bars plus site means and standard deviations on this plot. There is a lot of discussion of 'average' values and ranges and so the descriptive statistics should be presented.

Paragraph 307-318 – I've misunderstood something here: I couldn't follow the description of the dataset in this paragraph against Fig 2, which is the only figure cited. It seems to suggest that Fm values at the different sites are variously similar or different but in ways I couldn't see. For example, Fm values at KIS and BIS are supposed to be similar but those at WIS and SLW different from the former. I disagree – there is only one value at BIS and that is similar to 2 of the three values at KIS but very different to the third value, and the range at KIS looks comparable to ranges at WIS and SLW, albeit with a very small n. If this is actually referring to model output, where is it illustrated? Apologies if I have missed something here.

I think the same section uses 'statistically independent' (which has a very specific meaning) when 'statistically distinguishable' might be what is actually meant.

**Comparison to climatic forcings:** Without seeing the uncertainty bounds of the results it is difficult to assess the robustness of the conclusions. To be clear I think the paper raises some really interesting questions about the driving factors for grounding line retreat but without a greater assessment of model output sensitivity and uncertainty on durations/dates it is difficult to comment firmly on the conclusions. Whilst I understand why the authors are drawing out differences to Kingslake et al I think it is important to note that Kinglake et al used an ice sheet model to explain forcing mechanisms and they found that they could not initiate readvance without including some sort of GIA processes or buttressing from ice rise formation. Their ice sheet model was forced by a similar temperature record to the one described in Fig 7 and so a simple comparison of timing of events probably needs to be tempered by a discussion of whether temperature changes would be sufficient to initiate readvance. I also note the use of Hall et al (2006) sea ice record as a proxy for ocean warming – there may be better records of Holocene ocean temperatures such as Cunningham et al (1999) (The Holocene) which are based on oceanic proxies alone (Hall et al note that their record may reflect both oceanic and atmospheric forcing).

**Some line-by-line issues:**

44 – this hasn't been a consensus for quite a while, starting probably with Bradley et al 2015 but also including Matsuoka et al 2015 (Earth Science Reviews) which discussed smaller-than-present configurations and the GIA and climatic mechanisms that would explain Crary ice rise etc.

58 – the comparison to Greenland is not as useful as it might be – the main part of the GrIS that retreated and readvanced in the Holocene is in the SW where it is mostly a terrestrial margin, and so unlike the WAIS ocean and GIA forcing would not be possible.

92 – "chose the freezing point of freshwater" (insertion of 'fresh' reduces confusion as to whether you were just sticking with seawater (line 83) after Phase 1). Also, is there an argument to look at a 3-phase model where the heat flux sets the basal temperature in a subsequent Phase 3 (a la Bindschadler) after a period of basal freeze-on

98 – 'retreated' rather than 'advanced' otherwise I don't think this makes sense

148 – not equation 7

240-244 – I'm not quite sure what this is describing – can you explain the inference that comes from the grounding line constraint of 8000 and why the preceding 4000 years is used.

242 – should this read 'earliest' rather than 'latest'?

293- add reference 14/12 value as a line to figure 2a.

297 – add ratio as a line to Fig 2c

298 – Fig 9 – called out of sequence (see comments above about reordering this figure)

308 – I don't see eight sample points – I see 8+3

325-327 – use of 'model' to refer to (I think) 2 or 3 different models. Would really help to specifically refer to radiocarbon, temperature or ionic models – I lost track of the logical thread here.

328 – do 'model matches' on axis of Fig 7 mean the same as 'positive models' here

323-324 – refer to Figs 7 and 8 together

404 – what is ultimate origin of terrestrial plants?

408 – the values of C:N at UC are lower than at WGZ

415 – Bradley et al 2015 give a series of observations that support retreat and readvance including observation of (unstable) readvance of grounding lines on reverse slopes, amongst others.

**Figures and Tables**

**Figure 1**
I can't see a cyan diamond. Would be helpful to define abbrevations in the caption to save constantly referring back to the text.

**Figure 2**
add reference lines (see above)
Caption suggests C(org):N(org) but I think it should be  C(org):N(total).
Caption – '…matter from…..'
Caption suggests the C:N ratios of data in the paper are plotted as atom:atom but looking at the fields used from Lamb et al (2006) I believe they were plotted in the original paper as weight ratios. So the weight:weight column in Table S1 needs to be plotted, not atom:atom. I think this will lead to a shift of *1/1.17 for all your data points. All text, including caption, plus results and conclusions will need to be checked to see if the correction changes anything in text.

**Figure 3**
Labels for T(o) and T(i) should be centred over their durations on the diagram, not placed at their end points otherwise there is confusion that they are dates not durations.

**Figure 4.**
Thin colour lines not clearly discernible on top of background colour palette for temperature gradient

Not clear why there are 7 lines for 3 sites – I thought it was perhaps for different cores ? If so, please note this in caption (I couldn't find how many cores there were at UC).

It would be very helpful to give the observed values in a table somewhere – otherwise it relies on ability to read off a contour from these colour plots. This would also allow the uncertainties in the observed gradients to be provided.

**Figure 5.**

y-axis caption – not clear. Number of what ? (runs ?) Overlapping with what ? (presumably within 2-sigma 'match' to observations ?) Is this number or a percentage ? (looking at the next figures you use percentages)

**Figure 6.**

These are quite rich diagrams and I can see the potential but it would be very helpful to include a caption that walks the reader through interpretation. E.g. "The key shows the percentage of xx model runs that………….Fig 6c shows that at KIS, ocean exposure durations, T(o) of approx. 1200-2200 yrs are preferred by the model but the model provides less constraint on preferred grounding line readvance duration, T(i)"

756 - I can't follow what this means – produced *what* simulated values of Fm and TOC ? (presumably all the runs produced simulated values of these two variables?). And what do you mean by 'fit measured values' – within 2-sigma ?  Some other measure? How many runs of the model do these % correspond to?

**Figure 7**

Worth reversing x-axis so that time goes same way as in rest of paper.

What are the thin curves plotting ? – I'm presuming it is the optimum ('highest percentage') values of To and Ti taken from Fig 6? How does the stacking work ? The results in fig 6 look like they are combined results for each site already.

These curves in Fig 7 also need some expression of uncertainty on them – for example Fig 6b shows that the curve for WIS should have broader uncertainty ranges than the curves for KIS in Fig 6c but at present this is not reflected in Fig 7.

Explain which constraint is plotted in the solid bars for T(i) – is it the temperature modelling?

As with figure 6 – the caption could be more helpful to a reader – it took me some time to work out that (I think) it is the peaks in the thin curves that I am looking at for most likely exposure-readvance duration combinations and which therefore are the most likely date for initial retreat past the site?

**Figure 8**

The blue and orange colour bars for frequency of successful runs have no scale.

Cyan diamond not visible to me

Add WIS and BIS/KIS flowlines to panel b.

**Figure 9**

See notes above. ND not needed in axis label for x-axis

**Supplementary**

77 – Celsius

Eqn S14 – should this be m(b) rather than m?

147 – if your model cannot distinguish between the two conceptual models then should say so

155 – the reference here to provenance relates to flowline (and geology) differences rather than any differences in timing behaviour and so I don't think it is relevant

Table S1

units missing from several columns

Make core nomenclature clearer e.g. in first row is this the data for RISP Core 7 at a depth of 46.6cm or for core 7-4 at 6.6 cm ?

---

## Short Comment (SC2) · 20 Mar 2021

Thank you very much for your comments. We appreciate how detailed they were. Addressing the issues you brought up will definitely make our manuscript stronger.

---

## Author Comment (AC1) · 3 May 2021

**General comments:**

Neuhaus and colleagues present a new modelling approach to constrain post-last glacial maximum grounding line behavior in the Ross Sea sector of West Antarctica. In this study, the authors explain previously published (Kingslake et al., 2018) radiocarbon data from subglacial and sub-ice-shelf sediment samples using a two-phase model for radiocarbon input and decay to determine the timing of grounding line retreat beyond sampling sites beneath Whillans, Kamb, and Bindschadler ice streams. The timing of re-advance over the same sites was determined using previously published basal temperature gradients (Engelhardt, 2004), porewater chemistry profiles (at Whillans Subglacial Lake; Michaud et al., 2016), and geophysical evidence (at the grounding zone of Whillans and Kamb ice streams; Horgan et al., 2013; 2017). Given that both the style (Swinging gate vs. saloon door vs. marine-based vs. retreat and re-advance; reviewed in Halberstadt et al., 2016) and timing of grounding line retreat in the Ross Sea Embayment (recently reviewed in Prothro et al., 2020) is the subject of active debate in the community, this re-assessment of previously published data has the potential to draw interest from both modern- and paleo-glaciological researchers. However, several points necessitate significant revision before this manuscript is accepted for publication.

**Major comments on scientific content:**

1. *Assumptions of the radiocarbon model: The main assumption of this study is that any radiocarbon present in subglacial sediment samples in this region comes from the marine environment. This assumption is supported by two previous studies—Kingslake et al. (2018), which is cited in the main text, and Venturelli et al. (2020) which is only cited in the figures. The authors should include further discussion of these two papers in the explanation of their assumptions to make it clear that this point is well-established in previously published literature. The model assumes that radiocarbon was added to these sediments at a constant rate while exposed to the marine waters (line 17-18 of supplement). This assumption should be justified with respect to the proposed mechanism of radiocarbon input to these sediments (228-229 of main text)—was radiocarbon addition set to a constant rate because it is physically realistic for fecal pellets and faunal necromass to be deposited at a constant rate or because it is the simplest way to model input? How much would a variable rate of radiocarbon input change the timing of grounding line retreat? What is the uncertainty that this assumption imposes on the result? In the independent evolution of $^{12}C$ in this model, further explanation of carbon supply should be included. There is reference to radiocarbon free material on continent with reference to a previous study (Tulaczyk et al., 1998); however the authors set carbon input to 0 after the grounding line re-advances. How would the transport of radiocarbon-free material by the overlying ice streams impact model outputs? A sensitivity test of variable vs. constant carbon inputs would demonstrate that this assumption is sound, similar to what was done for the ice temperature model.*
     1. *Authors' response:  Yes, we agree that it is established in the literature that the radiocarbon present in our subglacial sediments comes from a marine environment.  We have added a sentence to section 2.3 clarifying that this idea has been previously established.  We have also added further reference to, and discussion of, the Venturelli et al. (2020) paper throughout our manuscript.  We assumed that radiocarbon was added to the sediments at a constant rate for model simplicity.  However, we do examine a wide swath of accumulation rates in our model runs.  We like the reviewer's suggestion of sensitivity testing of variable carbon input (for both phases) and have added it to the supplemental.*

*Varying the rate of 14C and 12C during phase 1 and of 12C during phase 2 does not alter our results significantly.*

2. *Details about radiocarbon model: Whereas the authors do a good job of laying out equations used in this model and provide a graphical depiction of model outputs (figure 6, and S1), significant details of the modeling methods are missing, preventing me making a careful assessment of this key piece of the manuscript. The caption of figure 6 provides a percentage of simulated Fm values that match measured values, however nowhere in the figure caption or text is it stated how many model runs were performed. As written, the manuscript lacks explanations for why the (maximum) 4% match between modeled and measured Fm shown in figure six is significant. Does this mean the model only reproduced the measured values 4% of the time?*

    1. *Authors' response:  We agree that including the number of model runs is key information for the reader.  We have therefore added the total number of model runs for the radiocarbon model to the caption for the figure presenting the radiocarbon model results.  The percentages for our radiocarbon model results are small because we run a very large number of simulations.  For each combination of Ti and To, we run the model for different starting concentrations of 12C and different rates of accumulation for 12C and 14C.  We choose these ranges based on the results from running phase one of the radiocarbon model for RISP and WGZ (the two sub-ice shelf sites).  Thus, for every combination of Ti and To, we run the model roughly 30,000 times.  For combinations of Ti and To that produce hundreds of model matches, the percentage of model matches is still small.*

3. *Proposed mechanism of radiocarbon input:* The authors state that radiocarbon in these sediments "likely" comes from fecal pellets and faunal necromass. Have intact fecal pellets or diagnostically Holocene macro/micro/meiofaunal parts been observed in previous micropaleontological investigations of these sediments (e.g. Harwood et al., 1989; Scherer, 1991; Scherer et al., 1998; Coenen et al., 2019) or subsequent investigation herein? If micropaleontological evidence for Holocene particulate carbon input does not exist, it should be made clear that this mechanism is assumed, and the assumption should be defended in this discussion of this manuscript.

    1. *Authors' response: It is well established in the literature that carbon input to the sediments in a marine environment comes from fecal pellets and faunal necromass (Kingslake et al., 2018; Turner, 2015 [Progress in Oceanography]). The abundance of fauna living in the water column at WGZ supports the idea the carbon input to the sediments at our field sites came from fallen faunal necromass and fecal pellets.  To help support this in the manuscript, we have added citations from Kingslake et al. (2018) and Turner (2015).*

4. *Uncertainty in retreat ages:* In its current form, the manuscript lacks any quantification of the uncertainty in modeled retreat ages. The interpretation presented in this manuscript suggests grounding line retreat 4,000 years before present at Whillans Ice Stream and 2,000 years before present at Kamb and Bindschadler ice streams. Based on the figures 6 and S1, it seems that the presented values may be the mean output of the radiocarbon model, however it is not clear from the methods or results if this is true. At the very least, it would benefit the manuscript to provide quantitative uncertainty of model outputs. I note that a "frequency of successful model runs" is included in figure eight, but there are no values tied to the color bars. It is therefore impossible to assess the weight of these results or contextualize them with previous studies. Given the noted assumptions in both the radiocarbon and temperature models, a full propagation of uncertainty should be presented to support the presented retreat and re-advance timing. Error bars should be

added to the presentation of timing in figures 7 and 8 and discussion should be added in text about whether or not the modeled timing in this study falls within error or previous studies from this region (Kingslake et al., 2018; Venturelli et al., 2020). The authors are specifically using their modeled retreat and re-advance timing to designate a climate-related forcing mechanism instead of previously suggested sea level and glacioisostatic forcings, but without any presentation of uncertainty it is unclear how reliable this alternative explanation is.

      1. *Authors' response: We have added in error bars to the timing of grounding line retreat and re-advance. We include those error bars in the figures where it is relevant, and include them in the text when we present those numbers.*

5. *Interpretation of geochemical data:* It is well-established that the subglacial sediments in this study contain a mixture of past marine and terrestrial microfossils (e.g. Coenen et al., 2019). Though this point is acknowledged in this manuscript (lines 169 and 301), the authors use bulk geochemical analyses (C:N ratios, Fm, δ13C) to make interpretations about the origin of organic material in the samples herein. At present, the manuscript lacks sufficient discussion of how a multi-source mixture would appear in the results of these geochemical analyses. Furthermore, the assertion of a marine source of organic material is not consistent with the shaded boxes in figure 9. It would benefit the explanation of new data generated herein to compare measured δ13C values to δ13C values for particulate organic carbon and sedimentary organic carbon in the contemporary Ross Sea (e.g. Villinski et al., 2000). This information would improve the presentation of data in figure 9, and further explanation should also be included in the discussion.

      1. *Authors' response: Yes, the subglacial sediments contain a mixture of pre-glacial terrestrial input and marine input. Based on the on the location of the subglacial samples in the d13C vs. C:N plot, we surmise that the majority of the organic material derives from the terrestrial sediment and only a small amount comes from marine sources. We have added a sentence clarifying this in section 4.2. We agree that there is room for further discussion of d13C and are working on analyses for this.*

6. *Temperature model:* In the current form, the manuscript lacks quantification for the uncertainty of re-advance timing. The sensitivity testing (Supplement section 2) indicates that the authors considered uncertainty in assumptions of this model and may therefore be more reliable than the retreat timing determined with the radiocarbon model. The constraint on re-advance timing has wide ranging importance from providing a constraint on Holocene ice dynamics to aiding in the interpretation of microbial data—a point that is very well illustrated in paragraphs on lines 372-394. However, the significance of this important result is overshadowed by a lack of error bars.

      1. *Authors' response: We have calculated error for the temperature analysis and now present it in the relevant figures. We also include the error bars in the text whenever we present our results for timing of grounding line re-advance.*

Uncertainty must be included for both the timing of retreat and re-advance to assess what new knowledge is presented in this manuscript. Without an explanation of the uncertainty in these age ranges, it is impossible to determine if modeled results are significantly different spatially (i.e., the difference between Whillans and Kamb/Bindschadler ice streams herein), temporally (i.e., Is there any overlap in the modeled timing of retreat and readvance at any site?), or from the many studies surrounding the chronology of grounding line retreat in this region (most notable to the sites included in this study: Spector et al., 2017; Kingslake et al., 2018; Venturelli et al., 2020). A discussion of all of these points must be included to set this study apart from previous work in this region. If the timing presented in this paper is significantly statistically

different than previous studies in this region, the inclusion of uncertainty has the potential to make this paper impactful for a wide-ranging scientific audience.

**Line/Technical comments:**

43: Smith et al., 2019 (doi: 10.1038/s41467-019-13496-5) provided a comprehensive review on this topic and should be cited here.

Author's response: Thank you for bringing this paper to our attention. We agree that this paper address the lack of data from below ice shelves and the need for more in order to provide more comprehensive grounding line retreat histories, and therefore should be cited here. We have thus included it.

43: You can/should state that this is the enduring paradigm in the Ross Sea Sector, and that this has been recently challenged [Bradley et al., 2015; Kingslake et al., 2018; Venturelli et al., 2020 (you could even add in Greenwood et al., 2018 if you wanted to include the EAIS portion of the Ross Sea Embayment)]. However, "scientific consensus" is too strongly phrased given the large body of work detailing the active debate in this region. As written, this also contradictory to the "disagreements" discussed in section 4 of the supplement. Some comprehensive reviews (e.g. Prothro et al., 2020 and Halberstadt et al., 2016) of this debate are not cited here or throughout the manuscript, but should be to provide better context on the state of knowledge.

Authors' Response: We agree that "scientific consensus" is too strongly worded. We have re-worded this paragraph to better reflect the evolution of ideas about post-LGM grounding line positions in the Ross Sea. We have also included the suggested citations.

Additionally, we have decided to modify section 4 for the supplement (saloon door vs. swinging gate) and move it into the main text. While we still examine the compatibility of our results to the swinging gate and saloon door models, we now also include a discussion of the role that bathymetry may have played on the grounding line retreat in the Ross Sea. We extend the conclusions of Halberstadt et al. (2016) and Prothro et al. (2020) that the grounding line retreated first in the troughs to the area currently under the Ross Ice Shelf. Bathymetry under the Ross Ice Shelf shows a large trough alongside the Transantarctic Mountains. We surmise that this trough aided the rapid retreat

63-66: There should be some detail added about why you conjecture re-grounding as the explanation for observed unsteady thermal state. Other processes that may explain this observation should be mentioned in text or even tested with your temperature model. A demonstration of how you ruled out these other processes would add strength to the statement in this sentence.

Authors' Response: We agree that this section deserves an explanation as to why we suspect that the unsteady thermal state is due to the grounding of an ice sheet. The high basal temperature gradients observed at KIS, BIS, and UC indicate that there is cold ice present at the base of the ice. The way to get cold ice at the base of the ice is by either lowering the surface temperature, increasing advection (either vertical or horizontal) or by changing the basal boundary conditions. Engelhardt (2004) ruled out lowering of the surface temperature and increased vertical advection (accumulation), settling on a combination of increased horizontal advection and basal melt. However, he required an extremely large increase in horizontal

advection and basal melt in order to reproduce the observed temperature gradients.  Thus, we believe our explanation of ice sheet grounding (i.e. changing basal boundary conditions) to be more realistic.  We have added in this explanation to the manuscript to rule out other processes and justify our choice of boundary conditions.

75-77: Do you plan to include your MATLAB code in the supplement? The reference here and elsewhere in the manuscript make me curious to see it.

Authors' response:  We can upload the MATLAB code to GitHub.  We are working on cleaning up the code to make it presentable.

87: Explanation for the assumed ice shelf thicknesses (500-1000 m) is needed. If these are based on the modern thickness distribution of Ross Ice Shelf, a citation should be included.

Authors' Response:  Yes, we assumed these ice shelf thicknesses based on the modern ice thickness at the Ross Ice Shelf grounding line.  We have added in a statement clarifying this.

91-93: Explain why you assume basal freeze on to have occurred after re-grounding. Do you have any age constraint on accretion? Justification for this assumption should be added.

Authors' Response: We do not have age constraints on the basal ice, but we assume that when this ice was part of an ice shelf there was abundant basal melting.  Begeman et al. (2018) and Marsh et al. (2016) presented basal melt rate measurements at the grounding zone of the Whillans Ice Stream that showed that there are very high rates of basal melt near the grounding line.  Thus, we surmise that any basal ice that may have existed prior to grounding line retreat melted away when the ice was floating.  Because there is basal ice present at our field sites now, we conclude that it must have frozen after the sites re-grounded.  We agree that an explanation of this is needed, and have thus included one in the manuscript.

95-99: It would benefit this manuscript to add a brief explanation of why you set 10,000 years as the model boundary. I realize that this is tied to the work of Kingslake et al. (2018), but it would improve the clarity of this paper to add a sentence or two to explain this so the reader does not have to search through another paper for this explanation.

Authors' Response: We originally chose to examine the grounding line position over the past 10,000 years because Kingslake et al. (2018) modelled the grounding line retreat over our field sites 9.7 kya.  However, we have since decided to alter that time window to cover the past 8000 years rather than 10,000 years because we wanted to use grounding lines which have been interpreted from measurements rather than models (Lee et al., 2017; McKay et al., 2016; Spector et al., 2017).  These studies place the grounding line several kilometers north of our field sites ca. 8000 years ago, thus we believe that the grounding line could not have retreated over our field sites earlier than 8000 years ago.  We agree that the reader deserves an explanation that does not require them to search through another paper.  We have therefore changed the date from 10,000 years to 8000 years, and included a more thorough description of why we chose this date.

98: Should "advanced" be swapped for "retreated" here?

Author's response: Yes, this should read "retreated" as opposed to "advanced." We appreciate the reviewer for pointing out this mistake. We have corrected it in the manuscript.

102: Here and throughout, "Subglacial Lake Whillans" should be changed to "Whillans Subglacial Lake" to align with the official place name established in 2018. Further details can be found with the direct link provided below:
https://geonames.usgs.gov/apex/f?p=138:3:::NO::P3 ANTAR ID,P3 TITLE:19707,Whillans%20Subglacial%20Lake

Author's response: Thank you for pointing out to us the correct naming convention for Whillans Subglacial Lake. We have changed all instances of "Subglacial Lake Whillans" to "Whillans Subglacial Lake" to reflect the proper title.

136, 144: Provide further explanation for the instantaneous change in boundary conditions. Do you assume that there is no transitional phase in which seawater and subglacial water are mixed as the grounding line re-advances, as has been proposed for grounding lines in this region? (e.g. Horgan et al., 2013 doi: 10.1130/G34654.1)

Author's response: We thank the reviewer for brining to our attention the paper on estuarine conditions below the Whillans ice Stream. We agree that the transition between sub-ice shelf and subglacial conditions is more nuanced than represented in our model. Due to tidal pumping, marine waters are actually able to enter the subglacial system upstream of the grounding line. However, for convenience, we chose to represent this transition as instantaneous in our model. We have added a statement to this effect in the manuscript for clarification.

169: "Meaningless" should be changed to "Chronologically meaningless". The presence of radiocarbon in subglacial sediments was used in Kingslake et al (2018) to challenge the enduring paradigm of grounding line retreat in this region. That alone makes those data meaningful, and your work in this manuscript has the potential to add further value.

Author's response: The reviewer makes a good point here. The radiocarbon ages are meaningful, even though Kingslake et al. (2018) were not able to use them to constrain the timing of grounding line movement. We have changed the wording so that we do not diminish the importance of the age results.

176: Sediments intended for geochemical analyses of acid insoluble organic material are conventionally decarbonated using hydrochloric acid. Some explanation should be added about why a different method is used here. Important details about the size of samples (mg? g?) decarbonated using this method are noticeably absent.

Authors' Response: TOC was estimated directly with C-EA-iRMS instead of indirectly by difference (TOC=TC-TIC). To prepare a bulk sediment sample for direct TOC measurement carbonate extraction, the higher pH of sulfurous acid (pH=4.5) is more specific than hydrochloric acid (pH<1). Carbonate is digested at pH 4 to 5, and at its very low pH, hydrochloric acid may also digest some of the organic material we are trying to measure. Thus, sulfurous acid was used instead.

The sample size used to make the measurements was 8-10 mg of bulk sediment. We have added in this piece of information to help clarify our methods.

225: Space between number and unit (100 g)

Author's response: Thank you for pointing this typo out. We have added the space.

228-229: I make note of my questions surrounding your radiocarbon input mechanism above (#3), but the statement about advection here raises further question. Are you assuming that particulate carbon is being advected under Ross Ice Shelf from the open marine environment or elsewhere in the sub-ice-shelf environment?

Authors' response: Ultimately the radiocarbon-bearing organic matter comes from the open marine environment in front of the Ross Ice Shelf. Organic matter originating from the subglacial environment contains radiocarbon-dead matter.

307: Provide explanation when stating that something is "surprising". If the primary source of sedimentation at your sub-ice shelf sites is melting of basal debris from the overlying ice shelf that you assume accreted following grounding line re-advance, one could expect these samples to be geochemically similar.

Author's response: If one came to this manuscript with the mindset that the sites which are currently subglacial have been so since before the LGM, it would be surprising to find that these sites have similar Fm values to sites which are currently located in the ocean. However, given that the idea that the grounding line retreated over these sites during the Holocene is now established in the literature (Kingslake et al., 2018; Venturelli et al., 2020) this is perhaps less surprising. We will therefore remove the word "surprisingly."

334-340: This paragraph makes it seem like you are assuming that sediments present at the surface when your samples were collected are the same sediments that were present at the surface when grounding line retreat occurred. If I am interpreting your assumption correctly, provide some context for why significant sediment accretion would not have occurred in the ~2000-4000 years between your modeled retreat and sample collection.

Authors' response: Hodson et al. (2016) found no evidence of sediment erosion or deposition at this field site (SLW). Additionally, there has not been thorough vertical mixing of the subglacial till layer due, in part, to the presence of a shallow lake preventing contact between the sediments and the ice, and to the fact that Ti is short. Thus, we expect that the topmost sediments at SLW were deposited when the site was exposed to the ocean. We have added in a sentence to the manuscript clarifying this.

365: Simkins et al., 2018 (doi: 10.5194/tc-12-2707-2018) would be a good citation to strengthen the grounding zone wedge sentence here.

Author's response: We agree that this paper strengthens our argument that GZWs can form rapidly (i.e. multi-year to decadal time scales) and have thus cited it here. Thank you for bringing this paper to our attention.

403-405: Should add an explanation of the proportion of inputs needed to result in the values shown in figure 9.

Authors' response: Based on the position of the subglacial sediments on the d13C vs. C:N plot, we believe that the majority of the organic matter originates from pre-glacial terrestrial C3

plants. However, the sediments also contain a small amount of marine input that is responsible for the radiocarbon. Bu that input is small enough that it does not cause the samples to display a marine signature in the plot. We have added a sentence that make it clear that the marine input is small relative to the pre-glacial terrestrial input.

Figure 2: Error bars for geochemical data (Fm, %TOC, C:N) are needed (and should be propagated through all analyses).

Author's response: We have added error bars to the plots of TOC and C:N. We did not include the error bars for the measurements of Fm because they are smaller than the symbols, and we made note of this in the figure caption. We had accounted for the error in Fm in our analyses, but had not accounted for error in TOC measurements. We therefore re-ran our radiocarbon simulations. The changes in the results were very slight (i.e. did not alter our conclusions) but we did remake Figures 6, 7, and S1 so that they were in keeping with our findings.

Figure 4: Note how many temperature model runs were performed.

Author's response: The total number of temperature model runs performed was 808,000. We have added this number to the figure caption to give the reader more information about our methods.

Figure 5: Are these results for ionic diffusion modeling efforts of all chemical parameters noted in Table 1? How many model runs were performed?

Author's response: In Figure 5 are shown the stacked results of the ionic diffusion modelling of all six chemical parameters noted in Table 1. The total number of model runs used to create this figure was 19,926. We have added this information to the figure caption because we agree with the reviewer that this information is important for the reader to better understand our results.

Figure 6: It is stated in the caption that the model runs are stacked for each core, but it is not indicated how many model runs were performed.

Authors' response: For each core, a total of 103,495,644 model runs were performed. To obtain the number of model runs shown for each field site, one must multiply that number by the number of cores located at each field site. We have added these numbers to the figure caption to provide further context.

Figure 7: Add uncertainties for timing indicated by colored lines. Are they within error of measured values of Venturelli et al (maroon error)? It would be interesting to supplement this point in your figure with some explanation in the discussion.

Authors' response: We have added uncertainties to our estimates of timing of both grounding line retreat and re-advance. Our estimates of the timing of grounding line retreat over SLW and WIS line up nicely with the estimates of grounding line retreat over WGZ from Venturelli et al. (2020). There is some overlap in those timings, although generally it would appear that the grounding line retreated over WGZ earlier than SLW or WIS, which is to be expected as it is over 100 km upstream of those sites. We agree that this is a useful observation, and have thus added it to the discussion of timing of grounding line retreat in section 4.1.

Figure 8: Do the color bars for frequency of successful model runs in 8a have any associated number values? Explanation should be included in the figure caption for what designates a successful model run for grounding line retreat. Figure clarity would be improved by adding a key for the shape points and line colors in 8a rather than an explanation in the caption.

Authors' Response:  The color bars are simply the probability density plots for timing of grounding line retreat seen in Figure 7.  We have added this information to the caption to clarify what constitutes a successful model run.  We have also added numbers to the key so that the frequency of successful model runs can be more easily discerned.  We also recognize that the term "frequency of successful model runs" might be confusing because we have been using "model matches" or "positive model matches" throughout the manuscript.  We have therefore changed the wording in the figure to "percentage of model matches for grounding line retreat."

Given the number of symbols used in panel a, we find that adding a legend actually makes the figure too busy.  We therefore have decided to keep the description of the symbols in the figure caption.

Figure 9: The x-axis label says $C_{org}/N_{tot}$, but the caption says $C_{org}:N_{org}$ It seems from the supplement and methods that only Total Nitrogen (TN) was measured. Can you clarify?

Author's response: Thank you for pointing out this inconsistency.  The figure caption is incorrect. We plotted $\delta^{13}C$ vs. $C_{org}:N_{tot}$ in Figure 9.  We have corrected the figure caption to reflect this.

Supplement 143-144: The "Saloon Door" comes from Ackert (2008). This citation should be added in addition to the already included citations for later discussions of this model.

Authors' Response:  Thank you for providing us with this citation.  We have included it to this paragraph.  Additionally, we have expanded the discussion to include discussion of the findings from Halberstat et al. (2016) and Prothro et al. (2020).  Because of these changes, we feel that this section is relevant to our manuscript and have therefore moved it to the main text.

**Below are citations included in this review that are not already included in the manuscript:**

Ackert, R. (2008, October). Swinging gate or Saloon doors: Do we need a new model of Ross Sea

deglaciation. In *Fifteenth West Antarctic Ice Sheet Meeting, Sterling, Virginia* (Vol. 811).

Coenen, J. J., Scherer, R. P., Baudoin, P., Warny, S., Castañeda, I. S., & Askin, R. (2020). Paleogene

marine and terrestrial development of the West Antarctic Rift System. *Geophysical Research*

*Letters*, *47*(3), e2019GL085281.

Halberstadt, A. R. W., Simkins, L. M., Greenwood, S. L., & Anderson, J. B. (2016). Past ice-sheet

behaviour: retreat scenarios and changing controls in the Ross Sea, Antarctica. *The Cryosphere*, *10*,

1003-1020.

Horgan, H. J., Alley, R. B., Christianson, K., Jacobel, R. W., Anandakrishnan, S., Muto, A., ... & Siegfried,

M. R. (2013). Estuaries beneath ice sheets. *Geology*, *41*(11), 1159-1162.

Prothro, L. O., Majewski, W., Yokoyama, Y., Simkins, L. M., Anderson, J. B., Yamane, M., ... & Ohkouchi,

N. (2020). Timing and pathways of East Antarctic Ice Sheet retreat. *Quaternary Science Reviews*, *230*,

106166.

Simkins, L. M., Greenwood, S. L., & Anderson, J. B. (2018). Diagnosing ice sheet grounding line stability

from landform morphology. *The Cryosphere*, *12*, 2707-2726.

Smith, J. A., Graham, A. G., Post, A. L., Hillenbrand, C. D., Bart, P. J., & Powell, R. D. (2019). The marine

geological imprint of Antarctic ice shelves. *Nature Communications*, *10*(1), 1-16.

Villinski, J. C., Dunbar, R. B., & Mucciarone, D. A. (2000). Carbon 13/Carbon 12 ratios of sedimentary
organic matter from the Ross Sea, Antarctica: A record of phytoplankton bloom dynamics. Journal of
Geophysical Research: Oceans, 105(C6), 14163-14172.

Author's Note:  We would like to thank the reviewer for a very thorough review, and for providing excellent feedback.  We also appreciate the recommendations for citations.

---

## Author Comment (AC2) · 3 May 2021

**Review of Neuhaus et al – Did Holocene climate changes drive West Antarctic grounding line retreat and re-advance?**

Firstly, my apologies that this review is right up against the deadline that I was given – I had some unexpected personal circumstances.

Secondly, for contextualising some of my several comments on figures it is worth noting that I have a mild but common colour deficiency. It might be helpful for the authors to know that as a general rule when small colour symbols (or thin lines) are placed on top of other colours it can be extremely difficult to distinguish the colours and match them to a key. There is now wide debate on this in scicomms literature and some good reviews available to guide authors e.g. https://www.climate-lab-book.ac.uk/2014/end-of-the-rainbow/ – most of R, GMT, MATLAB etc now have colour-vision- friendly palettes available.

This paper aims to address the question of timing of a grounding line retreat and re-advance in the Ross Sea sector. This has been part of a debate for some time on retreat history of the region but this paper particularly follows on from the work of Kingslake et al (2018) which attempted to constrain retreat timing but which yielded dates for retreat that were inconsistent with significant amounts of other observations.

In this paper, the authors take three main approaches to constraining past timing of retreat and readvance: and in each case comparing model simulations to archived (and some new) measurements from subglacial and sub-ice shelf sample sites. First, they use temperature modelling of the basal ice to constrain the timing of grounding line readvance. Secondly they use iconic diffusion modelling of the topmost portion of subglacial sediment to constrain timing of readvance. And thirdly they use modelling of radiocarbon content of subglacial sediment to constrain the period of ocean exposure between grounding line retreat and readvance. The authors then go on to draw some conclusions on the likey forcing factors and conclude that the drivers for retreat and readvance were climatic rather than delayed ice dynamics and glacioisostatic adjustment, as suggested by Kingslake et al.

The paper is based on an interesting idea which has potential to provide insight into the grounding line history, and I very much like the use of archived samples and long-standing measurements. The results will be potentially of interest to a broad community, but they need further explanation and some additional clarity (especially on uncertainty) before the paper should be accepted for publication.

**Broad issues**
**Model details.** The derivation of model equations is mostly well dealt with but some other details such as numbers of runs, uncertainty (see below), and how to interpret some of the output (especially in some very rich figures), are not yet clear enough.

*Authors' Note: We appreciate the feedback on interpreting the figures. We recognize that some of our figures contain a lot of information and we have added further explanations in the captions to try and assist the reader in interpreting them. We have also added information about the number of model runs to the relevant figures (Figures 4, 5, and 6). We have also calculated the uncertainty in the timing of grounding line retreat and not only added them to the relevant figures, but added them to the text whenever we discuss the timing of grounding line retreat or re-advance.*

**Model assumptions** – in a number of places it is difficult to follow if assumptions/choices are made for model simplicity or are based on a comprehensive observational dataset. For example, a single porosity value of 0.4 (line 126) seems surprising: marine sediments vary significantly in porosity and I imagine that subglacial sediments do too. Similarly the supplementary gives a single value for geothermal heat flux but without discussing any likely range or sensitivity of the model to this parametrisation. Whilst the choice of parameters may be helpful it is difficult to assess the model results without knowing the uncertainty bounds created by uncertainty in parametrisation. Some sort of sensitivity or uncertainty analysis of the model is needed to try and understand the range in final exposure/readvance durations. For example, the plots in Fig 6, which I read as a form of probability density plot translate into single thin lines on figure 7 but without uncertainty bounds included.

*Authors' Note: In our models we chose parameters based on a combination of observed and assumed constraints. We will adjust our language to make it more clear which are which. We have also performed sensitivity tests of the models and have found that changing our assumptions does not alter our results that much. We are including these sensitivity tests in the supplemental.*

**Unmodelled processes:** What would be the effect of sediment accretion and/or deformation following grounding line readvance? Given this is the 'type area' for actively deforming till layers it probably needs some qualitative comment on the likely effects of such deformation on the radiocarbon and/or ionic diffusion observations. If it is only simple shear then there would be no vertical movement but till accretion is possible and this would likely lead to addition of deep interior radiocarbon-dead carbon. I think some qualitative discussion on possible post-depositional changes would be important here, as the implicit assumption is that the sediment being sampled was the same sediment exposed as the cavity was closed.

*Authors' Note: We agree that our manuscript would benefit from discussion of the effects of sediment accretion and/or deformation. We are working on including this. We have performed sensitivity testing of the radiocarbon model where we allowed sediment deposition (in the form of $^{12}$C) to occur during phase 2 (grounding line re-advance). This did not alter the results of the radiocarbon modelling noticeably. We are adding this sensitivity testing to the supplemental.*

**C:N rationale and treatment:** The explanation of why C:N is being measured and analysed does not become clear until the Results in Section 3.3, when Figure 9 is called (out of sequence) and the different fields for marine and terrestrial become clear. I suggest putting Fig 9 much earlier and calling it from the methods where it will be possible to explain the concepts behind measuring C:N ratios and del-13C. This explanation can also then make clear the reasoning as to why terrestrial plants field matter (I presume this is because the radiocarbon-dead material is assumed to come from long-dead terrestrial plant material in bedrock beneath the WAIS but I couldn't see it made explicit anywhere). Note also comments below on use of weight:weight and atom:atom ratios on the same plot.

*Authors' Note: Thank you for pointing out that we called Figure 9 out of sequence. We like the suggestion of referring to Figure 9 in the methods as a way to clarify why we're interested in d13C and C:N measurements. We have therefore included a few sentences in the paragraph where we first mention the d13C and C:N measurements that explain how the relationship between d13C and C:N can be used to clarify the origin of the sediments. Incidentally, we do mention in section 3.3 that the sediments consist of a mixture of marine organic matter (which is responsible for the radiocarbon) and*

*dead C3 plant matter left over from before the continent was glaciated. Because this was not clear enough, we added a line to section 4.2 where we specify that the bulk of the organic matter in the sediments comes from the radiocarbon dead terrestrial matter, which explains the location of the subglacial samples on the d13C C:N plot. The marine matter is important because it contains the radiocarbon, but it is makes up a relatively small proportion of the total organic matter.*

*There was some confusion as to the units of C:N throughout the manuscript. We have cleared this up so that all values are now presented in atom:atom (see comments to Figure 2).*

**Presentation of (radio)carbon data on sediments**: It would be helpful to include analytical error bars plus site means and standard deviations on this plot. There is a lot of discussion of 'average' values and ranges and so the descriptive statistics should be presented.

*Authors' Note: We have added error bars to Figure 7 indicating the extent of our uncertainties. Additionally, in the text we now include the error bars whenever we talk about the timing of grounding line retreat or re-advance.*

Paragraph 307-318 – I've misunderstood something here: I couldn't follow the description of the dataset in this paragraph against Fig 2, which is the only figure cited. It seems to suggest that Fm values at the different sites are variously similar or different but in ways I couldn't see. For example, Fm values at KIS and BIS are supposed to be similar but those at WIS and SLW different from the former. I disagree – there is only one value at BIS and that is similar to 2 of the three values at KIS but very different to the third value, and the range at KIS looks comparable to ranges at WIS and SLW, albeit with a very small n. If this is actually referring to model output, where is it illustrated? Apologies if I have missed something here.

*Authors' Note: We made a mistake when plotting Figure 2a, which accounts for the confusion here. Our analysis in the text was correct, and that can now be more easily understood by looking at Figure 2a. We appreciate the reviewer for bringing this to our attention so that we could fix it.*

I think the same section uses 'statistically independent' (which has a very specific meaning) when 'statistically distinguishable' might be what is actually meant.

*Authors' Note: Yes, the reviewer is correct. This should be 'statistically distinguishable' rather than 'statistically independent.' We have corrected that in the text.*

**Comparison to climatic forcings:** Without seeing the uncertainty bounds of the results it is difficult to assess the robustness of the conclusions. To be clear I think the paper raises some really interesting questions about the driving factors for grounding line retreat but without a greater assessment of model output sensitivity and uncertainty on durations/dates it is difficult to comment firmly on the conclusions. Whilst I understand why the authors are drawing out differences to Kingslake et al I think it is important to note that Kinglake et al used an ice sheet model to explain forcing mechanisms and they found that they could not initiate readvance without including some sort of GIA processes or buttressing from ice rise formation. Their ice sheet model was forced by a similar temperature record to the one described in Fig 7 and so a simple comparison of timing of events probably needs to be tempered by a discussion of whether temperature changes would be sufficient to initiate readvance. I also note the use

of Hall et al (2006) sea ice record as a proxy for ocean warming – there may be better records of Holocene ocean temperatures such as Cunningham et al (1999) (The Holocene) which are based on oceanic proxies alone (Hall et al note that their record may reflect both oceanic and atmospheric forcing).

*Authors' Note: We have now added error to the timing of grounding line retreat and re-advance.*

*Kingslake et al. (2018) were unable to re-create grounding line re-advance without GIA, although in their sensitivity testing they found that altering the ocean parameters had a large effect on the timing of grounding line retreat and re-advance. The results of Lowry et al. (2019) directly contradict Kingslake et al. (2018). Our results are more consistent with the results of Lowry et al. (2019), and dissecting why that is the case is beyond the scope of our paper. We are working towards helping to establish an observational basis for Holocene grounding line dynamics in the Ross Sea embayment. Our results disfavor the idea of Kingslake et al. (2018) that glacioisostatic adjustments were the most important to the Holocene dynamics of grounding lines in this region. If that were the case, then the re-advance should happen early in the Holocene when rebound rates were fast. Instead, our evidence points to alignment between grounding line retreat and re-advance phases with warm and cold (respectively) climatic phases observed in the regional paleoclimate proxies. We cannot comment on why different ice sheet models differ in their reconstruction of Holocene grounding line dynamics in our study region.*

*We appreciate the reviewer suggesting the Cunningham et al. (1999) study. We have included the bounds on their warm period in Figure 7 (which contains our results for the timing of grounding line retreat and re-advance) as well as discussing the timing of this warm period compared to our results. The grounding line retreat over SLW and WIS falls decidedly during their warm period in the mid-Holocene. The retreat over KIS and BIS coincides with that warm period within error. But the grounding line re-advance over all four sites falls within the cool period proposed by Cunningham et al (1999). We will still keep discussion of the two warm periods proposed by Hall et al, but we believe that our argument for oceanic influence on grounding line position Is strengthened by adding the Cunningham et al. (1999) paper to our discussion.*

**Some line-by-line issues:**

44 – this hasn't been a consensus for quite a while, starting probably with Bradley et al 2015 but also including Matsuoka et al 2015 (Earth Science Reviews) which discussed smaller-than-present configurations and the GIA and climatic mechanisms that would explain Crary ice rise etc.

*Authors' Note: We agree that it is too strong a statement to say that the idea of unidirectional post-LGM retreat was the consensus until 2018. We intended to convey that the idea of post-LGM grounding line retreat and re-advance has only been in the literature relatively recently (within the past 5-6 years). We have re-worded the paragraph to make this more clear. We have also added in the suggested citation. Thank you.*

58 – the comparison to Greenland is not as useful as it might be – the main part of the GrIS that retreated and readvanced in the Holocene is in the SW where it is mostly a terrestrial margin, and so unlike the WAIS ocean and GIA forcing would not be possible.

*Authors' Note:  We chose to mention the Greenland Ice Sheet in our introduction because there is already evidence that the Greenland Ice Sheet experienced Holocene fluctuations, which we are arguing WAIS did as well.  Irrespective of the exact mechanisms, both show sensitivity to the relative minor Holocene climate fluctuations.  Thus, we feel that this comparison is relevant to the introduction.*

92 – "chose the freezing point of freshwater" (insertion of 'fresh' reduces confusion as to whether you were just sticking with seawater (line 83) after Phase 1). Also, is there an argument to look at a 3-phase model where the heat flux sets the basal temperature in a subsequent Phase 3 (a la Bindschadler) after a period of basal freeze-on

*Authors' Note:  We agree that this should read "the freezing point of freshwater," to avoid any confusion that we might be examining the freezing point of ocean water.  We have thus made the change in the manuscript.*

*We did consider using the geothermal flux as a basal boundary condition, but we ultimately chose not to do this because we know there to be lots of water present under these sites.  Using the geothermal flux to constrain the basal temperature gradient only works when there is no water and ice is frozen to the bed.  Fisher et al. (2015) found that when water is present, the geothermal flux and the flux of heat through the base of the ice are not the same.  It would be difficult to recreate the steep observed basal temperature gradients using the geothermal flux as the basal boundary condition because it would quickly lower the basal temperature gradient.  The results using that basal boundary condition would indicate that the ice has been grounded for an even shorter period of time than we currently estimate.  Therefore, our results are actually a conservative estimate of the length of time the ice has been grounded.*

98 – 'retreated' rather than 'advanced' otherwise I don't think this makes sense

*Authors' note: Yes, this should read "retreated" as opposed to "advanced."  We appreciate the reviewer for pointing out this mistake.  We have corrected it in the manuscript.*

148 – not equation 7

*Authors' Note: Thank you for pointing out this inconsistency.  We actually intended to refer to equation 3.  We have made the correction so that we now refer to the correct equation.*

240-244 – I'm not quite sure what this is describing – can you explain the inference that comes from the grounding line constraint of 8000 and why the preceding 4000 years is used.

*Authors' Note: In order to constrain some of the variables used in the radiocarbon model (A and $N_0$) we ran the model for RISP and WGZ.  We needed to assume a time window over which the grounding line could have potentially retreated over WGZ and RISP.  We previously chose 10,000 years ago as the starting estimation based on the grounding line retreat model in Kingslake et al. (2018).  We then added 2000 year error bars to that time, which is how we ended up modelling the radiocarbon input at RISP and WGZ 8000-12000 years ago.  However, we have decided to change time window we examine to 8000 years ago to base the timing of grounding line retreat over RISP and WGZ on dates from Spector et al.*

*(2017) and McKay et al. (2016) which place the grounding line at Ross Island 8.6 kya and at Beardmore Glacier 8 kya. We chose to make this change so that our dates are based on geologic dating methods rather than modelling. Our previous time window did include 8000 years ago. We now will examine grounding line retreat over WGZ and RISP 8000 ± 1000 years ago. We have changed the wording in this paragraph to explain this more clearly.*

242 – should this read 'earliest' rather than 'latest'?

*Authors' Note: No, this should read "latest." We see how this could be confusing because 12,000 is a larger number than 8000. But 12,000 years ago is earlier than 8000 years ago. Thus, we will keep this as "latest."*

293- add reference 14/12 value as a line to figure 2a.

*Authors' note: By definition, the "modern" fraction of modern radiocarbon is 1. Fm is measured in reference to the modern ratio of $^{14}C$ to $^{12}C$. We have added a parenthetical clause clarifying this in the text. However, adding a reference line to Figure 2a would prove difficult as the measurements of Fm for our sites are very far from Fm=1, and extending the y-axis to include 1 would prevent us from seeing any variation on the measurements of Fm between our sites. Thus we have chosen not to include the reference line in Figure 2a.*

297 – add ratio as a line to Fig 2c

*Authors' note: We agree that it is useful to compare the C:N from our sites to the reference value for the ocean, and have thus added this value as a reference line in Figure 2c.*

298 – Fig 9 – called out of sequence (see comments above about reordering this figure)

*Authors' Note: We have moved Figure 9 as per the reviewer's suggestion. See earlier comment regarding the new ordering of this figure..*

308 – I don't see eight sample points – I see 8+3

*Authors' Note: Thank you for pointing out this mistake. First, there are in fact seven rather than eight total measurements of Fm from SLW and WIS. The mean and standard error calculations are correct, but we miscounted the number of points during the writing process. We have corrected this to read "seven" instead of "eight." Secondly, there was a mistake in the way that we plotted Figure 2a, resulting in more points appearing at WIS and SLW. This did not affect our analysis in any way, but we have corrected the figure so that the correct points are plotted.*

325-327 – use of 'model' to refer to (I think) 2 or 3 different models. Would really help to specifically refer to radiocarbon, temperature or ionic models – I lost track of the logical thread here.

*Authors' Note:  We agree that the fact that we refer to three separate models in this paragraph can be confusing.  We have therefore taken up your suggestion of specifically noting which model we are referring to throughout the paragraph.*

328 – do 'model matches' on axis of Fig 7 mean the same as 'positive models' here

*Authors' Note:  Yes, these are the same.  We have used 'positive model outcomes' and 'model matches,' which we realize is confusing.  We have therefore decided to just use 'model matches' for consistency.*

323-324 – refer to Figs 7 and 8 together

*Authors' Note: We are slightly confused about the line numbers presented here.  It does not make sense to include references to Figures 7 and 8 here as the conversation largely revolves around the half-life of radiocarbon and using the temperature and ionic diffusion models to constrain $T_i$.  However, if the reviewer is referring to lines 331-333, we agree that both Figures 7 and 8 contain relevant information and should therefore both be referred to.  We have therefore included the reference to Figure 7 as well as all of Figure 8 (rather than just Figure 8b) to lines 331-333.*

404 – what is ultimate origin of terrestrial plants?

*Authors' Note: The carbon that shows signatures of C3 plants is likely from tundra flora that was present before Antarctica was glaciated.  We would expect to see evidence of past C3 plants in Antarctica because they are typically found in cold, wet environments, much like Antarctica was before it was fully glaciated.  In fact, plant fragments and pollen have been found in sediment cores collected at our field sites (Coenen et al., 2019).   To convey this in the text, we specified that the C3 plants were pre-glacial.  We also added a sentence specifying that while the majority of the carbon in the sediments derives from dead C3 plant matter (which explains the position on the d13C vs. C:N plot), the radiocarbon comes from a small amount of marine matter which was deposited when the grounding line retreated.*

408 – the values of C:N at UC are lower than at WGZ

*Authors' Note:  The reviewer makes a very good point.  The C:N measurements from UC are, in fact, lower than those from WGZ.  It is likely that WGZ receives an input of subglacial sediments that cause the samples to have higher C:N ratios.  We have added a statement in the text clarifying that the UC samples have a lower C:N value than WGZ and that part of the explanation could be due to this subglacial input.*

415 – Bradley et al 2015 give a series of observations that support retreat and readvance including observation of (unstable) readvance of grounding lines on reverse slopes, amongst others.

*Authors' Note:  While Bradley et al. (2015) discuss observations that support the idea of recent grounding line re-advance, their study focuses on the Weddell Sea.  We were only considering the Ross Sea in the discussion in this paragraph, although we did not specify that clearly.  So discussion of the findings of Bradley et al. (2015) is not entirely relevant to this paragraph, although we do cite them elsewhere in the manuscript.  We have therefore added information letting the reader know that we are only discussing the Ross Sea region in this paragraph.*

**Figures and Tables**

**Figure 1**

I can't see a cyan diamond. Would be helpful to define abbrevations in the caption to save constantly referring back to the text.

*Authors' note: We apologize that the cyan diamond is not clearly visible.  We have changed the color to white and changed the shape to a star so that it is hopefully more visible.  We have also added the full names of the field sites to the caption as suggested.  Additionally, we changed color scale on the bathymetry so it more closely resembles the scale for the bathymetry shown in Figure 4a of Tinto et al. (2019).*

**Figure 2**

add reference lines (see above)

*Authors' Note:  We agree that adding a reference line to Figure 2c showing the typical ocean C:N is useful for putting our C:N measurements in context, and have thus included the line in Figure 2c. However, we have opted to not include a reference line in Figure 2a showing the modern standard Fm. The modern Fm is by definition 1, and extending the y-axis to include this would make it difficult to compare the values of Fm between the sites.*

Caption suggests C(org):N(org) but I think it should be C(org):N(total).

*Authors' note:  Thank you for pointing out this typo.  We have corrected it so that the caption now reads $C_{org}:N_{tot}$.*

Caption – '...matter from.....'

*Authors' note: We thank the reviewer for pointing out this typo.  We have corrected it.*

Caption suggests the C:N ratios of data in the paper are plotted as atom:atom but looking at the fields

used from Lamb et al (2006) I believe they were plotted in the original paper as weight ratios. So the weight:weight column in Table S1 needs to be plotted, not atom:atom. I think this will lead to a shift of *1/1.17 for all your data points. All text, including caption, plus results and conclusions will need to be checked to see if the correction changes anything in text.

*Authors' Note: There was some confusion with the numbers included in the table. The values of C:N for the RISP samples were atom:atom, but the values for the other samples were wt:wt. We have corrected this so that all values are shown in atom:atom. However, in Figure 9, all values of C:N were previously plotted as atom:atom as were the fields from Lamb et al. (2006). Thus the changes made to the table do not affect our results and conclusions.*

**Figure 3**

Labels for T(o) and T(i) should be centred over their durations on the diagram, not placed at their end points otherwise there is confusion that they are dates not durations.

*Authors' Note: We appreciate the reviewers suggestion of centering $T_i$ and $T_o$. We have added brackets above the diagrams so that it is more clear that $T_i$ and $T_o$ refer to durations of time.*

**Figure 4.**

Thin colour lines not clearly discernible on top of background colour palette for temperature gradient

Not clear why there are 7 lines for 3 sites – I thought it was perhaps for different cores ? If so, please note this in caption (I couldn't find how many cores there were at UC).
It would be very helpful to give the observed values in a table somewhere – otherwise it relies on ability to read off a contour from these colour plots. This would also allow the uncertainties in the observed gradients to be provided.

*Authors' Note: Thank you for pointing out that the lines are not easily discernible over the colormap showing the results of the temperature gradient. We decided to change the way that we present the results so that they are more easily understood by the reader. We now present histograms showing the distribution of the model matches (i.e. dates when the model output fell within 10% of the observed basal temperature gradients). We have also included a representation of the median and $32^{nd}$ - $68^{th}$ percentiles in the upper portion of the figure to better illustrate the uncertainty. Also, in the process of improving the presentation of results from the temperature model, we noticed a mistake which we have now corrected. Thus, the results from the temperature model are slightly altered.*

**Figure 5.**

y-axis caption – not clear. Number of what ? (runs ?) Overlapping with what ? (presumably within 2-sigma 'match' to observations ?) Is this number or a percentage ? (looking at the next figures you use percentages)

*Authors' Note: Thank you for pointing out that the label on the color bar is not clear. What we meant to convey with "Number of Overlapping 2σ" was that the color shown on the figure corresponded to the number of ionic profiles produced by the model that fit the observed ionic profiles within 2σ. We were stacking the results of six chemical parameters. What we were really trying to say was the "number of model matches." We agree that that is much less complicated to understand, and we have changed the label to "Number of Model Matches."*

**Figure 6.**

These are quite rich diagrams and I can see the potential but it would be very helpful to include a caption that walks the reader through interpretation. E.g. "The key shows the percentage of xx model runs that.............Fig 6c shows that at KIS, ocean exposure durations, T(o) of approx. 1200- 2200 yrs are preferred by the model but the model provides less constraint on preferred grounding line readvance duration, T(i)"

756 - I can't follow what this means – produced *what* simulated values of Fm and TOC ? (presumably all the runs produced simulated values of these two variables?). And what do you mean by 'fit measured values' – within 2-sigma ? Some other measure? How many runs of the model do these % correspond to?

*Authors' Note: We agree that these figures contain a large amount of information, and therefore deserve more explanation in the caption. We like your suggestion of walking the reader through the interpretation, and have thus included it in the caption. We have also added the total number of model runs performed for each field site to the caption. Finally, we have added the bounds on $T_i$ for each field site as determined from ionic and temperature diffusion modelling. This should help explain how we arrived at the curves shown in what was formerly Fig. 7.*

**Figure 7**

Worth reversing x-axis so that time goes same way as in rest of paper.
What are the thin curves plotting ? – I'm presuming it is the optimum ('highest percentage') values of To and Ti taken from Fig 6? How does the stacking work ? The results in fig 6 look like they are combined results for each site already.
These curves in Fig 7 also need some expression of uncertainty on them – for example Fig 6b shows that the curve for WIS should have broader uncertainty ranges than the curves for KIS in Fig 6c but at present this is not reflected in Fig 7.
Explain which constraint is plotted in the solid bars for T(i) – is it the temperature modelling?
As with figure 6 – the caption could be more helpful to a reader – it took me some time to work out that (I think) it is the peaks in the thin curves that I am looking at for most likely exposure-readvance duration combinations and which therefore are the most likely date for initial retreat past the site?

*Authors' Note: We agree with the reviewer that this figure contains a lot of information and could therefore be confusing without a clear figure caption guiding the reader to the interpretations. We have therefore re-worded the caption to be more descriptive. The colored curves are plotting the probability distribution of the results from the radiocarbon modelling. The difference between Fig. 6 and this figure is that we add $T_i$ and $T_o$ together and only examine the model results that fall within the error estimates for $T_i$ obtained through temperature and ionic diffusion modelling. To clarify which results we mean, we*

*have added dashed lines to Fig. 6.  What we were trying to convey by calling the curves "stacked" was that all cores are taken into account for each field site, but the reviewer is correct that we have already done that for Fig. 6.  Thus, we have removed that confusing statement in our re-wording of the figure caption.  We have also added error estimates to both the timing of grounding line retreat and grounding line re-advance.  We also added a statement clarifying that the estimates of $T_i$ presented in this figure come from the results of the ionic and temperature diffusion modelling.  Finally, the way that time has been presented on the x-axis in this plot is the same as in Figures 4-6.  We therefore think it makes sense to keep it in this configuration.*

**Figure 8**

The blue and orange colour bars for frequency of successful runs have no scale. Cyan diamond not visible to me
Add WIS and BIS/KIS flowlines to panel b.

*Authors' Note:  We apologize that the cyan diamond is not visible to you.  We have changed it to a white star to make it more visible.  The color bars correspond to the probability density plots in Figure 7.  We have added in percentages for context.  We are working on incorporating the flowlines from WIS and KIS/BIS to panel b.*

**Figure 9**

See notes above. ND not needed in axis label for x-axis

*Authors' Note:  We have removed [ND] from the x-axis label.*

**Supplementary**

77 – Celsius

*Authors' response: Thank you for pointing out this typo.  We have corrected it.*

Eqn S14 – should this be m(b) rather than m?

*Authors' response: No, in this instance "m" stands for "meter" and is part of the unit of measure [ºC/m]. It is not the variable $m_b$.*

147 – if your model cannot distinguish between the two conceptual models then should say so

*Authors' Note:  Yes, our results are compatible with both the "swinging gate" and "saloon door" models. We have added in a statement articulating this.  We have also decided to modify this section and move it to the main text because we feel that it is relevant to our discussion.  In addition to comparing our results to the swinging gate and saloon door models we include a discussion on the role that bathymetry plays on grounding line retreat.  We build on the conclusions made by Halberstadt et al. (2016) and Prothro et al. (2020) that the grounding line retreat initiated in the troughs and extend it below the current Ross Ice*

*Shelf. There is a large trough that runs along the Transantarctic Mountains, and we surmise that this could have aided the rapid grounding line retreat seen in Spector et al. (2017).*

155 – the reference here to provenance relates to flowline (and geology) differences rather than any differences in timing behaviour and so I don't think it is relevant

*Authors' Note: We agree that the reference to flowline and geologic differences is perhaps not relevant to this section. However, we have also slightly changed this section to include discussion of the sensitivity of grounding line retreat to bathymetry. In making those changes we removed this reference because we felt that it did not contribute to the conversation.*

Table S1
units missing from several columns

*Authors' response: Thank you for noting the omission. We have added units to the columns where the units were missing.*

Make core nomenclature clearer e.g. in first row is this the data for RISP Core 7 at a depth of 46.6cm or for core 7-4 at 6.6 cm ?

*Authors' note: We apologize for the confusion. We have reformatted the column and added dashes to make it more clear which numbers represent the core number and which represent the bounds on the section of the core.*

---

## Author Response (AR2)

Response to Reviewer 1:

In the revised manuscript entitled, "Did Holocene climate changes drive West Antarctic grounding line retreat and readvance", Neuhaus and colleagues have addressed most of the concerns presented in my first review. The authors have included uncertainties in their modeled chronological constraints on grounding line retreat and re-advance, along with significant differences in timing from the first version of this manuscript. As the objective of this manuscript is to shift away from the glaciological forcing presented by Kingslake et al. (2018) and toward a climate forcing, the timing and inclusion of uncertainties is very important. I note that though the peak model matches presented in the first version of this manuscript may be loosely correlated with modest changes in climate, a correlation between revised timing (with uncertainty) and changes in Holocene climate requires a more rigorous explanation than what is currently presented. My primary concerns remain:

1. Whillans Ice Stream is the only ice stream in the study with two sites along a flowline. This serves as an opportunity to test whether the model produces realistic results. I note that the upstream site (WIS) resulted in an earlier retreat time (4700 years ago) than the downstream site (SLW; 4300 years ago). Because this distribution of timing does not make physical sense, I am wondering if these were typed in backwards? Should SLW be 4700 years ago and upstream WIS be 4300 years ago? If this was not a mistake, this distribution of timing must be further explained. Additionally, there should be further discussion of why WIS and KIS/BIS sites result in significantly different retreat timing, as they are quite close together and would be experiencing similar changes in climate during the Holocene. The addition of Halberstadt et al. (2016)'s marine-based model indicates that the authors have considered differences in underlying geology/bathymetry. It should be stated clearly whether this is the assumed reason for differences in retreat.

Author's Note:  With regards to the relative timing of retreat at SLW and WIS sites, our method simply has too large uncertainties to engage in such analysis. The reviewer quotes just a single age for each of these two sites (4300 years for SLW and 4700 for WIS) without any error bars, but the same reviewer implored us previously to quantify uncertainties on these ages, which we did. These uncertainties are large and explain why these SLW and WIS model ages appear to be 'out of order'. Clearly, the grounding-line retreat must have happened first at SLW and then over WIS. The fact that the two ages do not appear to confirm that is just a reflection of the large uncertainties associated with these model ages. Therefore, we prefer to emphasize the entire probability distribution functions (PDFs) that are shown in our Figure 8. The two curves for WIS and SLW clearly show that the likelihood of retreat over SLW and WIS is similar over a wide range of ages.

There are at least a few reasons why the grounding line may have retreated over WIS/SLW before KIS/BIS. For instance, the grounding line of Bindschadler and Kamb ice streams may have been slower to retreat because of the buttressing effect of Roosevelt Island, where the grounding line was 'hung up' until 3200 years ago (which we mention in section 4.1). In addition, the deep bathymetric trough along the transantarctic mountains continues up through the Whillans Ice Stream whereas bathymetry beneath Kamb and Bindschadler ice streams is not as deep.  In section 4.3 we speculate that the grounding line retreats first in areas of deep bathymetry.  We have added in a sentence to section 4.3 spelling this out more clearly.

2. Do the results support a claim of a climate forcing? Retreat timings computed at BIS and KIS sites significantly overlap with the re-advance timings. The explanation for re-advance is slight cooling observed in the region over the last 2000 years, however, I note that the re-advance timing of these two sites falls closely in line with warm periods in the Ross Sea (Hall et al., 2006; grey shaded box in

figure 8) that the authors state forced retreat at WIS and SLW earlier in the Holocene. The full range of uncertainty for retreat encompasses 4000 years, a range that exceeds any of the associated changes in climate being used to explain the observed retreat and re-advance.

Author's Note: We fully disagree with this assertion of the reviewer. Yes, our estimates of grounding line movements in the study region are associated with considerable uncertainties. We have never tried to hide this fact. But what is the alternative? Is it better to have no estimates at all? Error bars of 4000 years are large but what were the error bars on the scientific understanding of grounding line retreat and readvance in the study region before our manuscript was submitted? It is easy to complain about uncertainties in other people's work but can the reviewer, or anyone else, offer a better alternative? It will most likely be decades, if ever, before anybody is able to collect samples and data that can improve on our estimates. Is a complete lack of constraints the better option? On our Figure 8 we marked with a maroon arrow the result of Venturelli et al. (2020) in which they have estimated that the grounding line of Whillans ice stream retreated past its modern location sometime in the mid Holocene. The uncertainty on their estimate is not much smaller than the 4000 years of uncertainty on our estimate. Yet, GRL reviewers and editors did not deem their result to be pointless, as this reviewer is asserting with regards to our results. Venturelli et al. (2020) concluded based on their results that the grounding line retreated during warm phases of Mid Holocene and then readvanced during Late Holocene. This published conclusion is remarkably like ours even though it is based on fewer samples from fewer locations. Frankly, we are completely at a loss how it is possible to look at our Figure 8 and to claim that we have no basis for hypothesizing that climate warming drove grounding line retreat and climate cooling caused the readvance in the study region. And we emphasize in the paper that this is a hypothesis or a conjecture. After all, even our title ends with a question mark. However much one can complain about the large size of our error bars, the fact is that the error bars for all four estimates of grounding line readvance overlap with the Late Holocene cooling seen clearly in the WAIS Divide record. And all four error bars for our estimates of grounding line retreat overlap with the warmest part of the WAIS Divide record shown in Figure 8. Since when did it become controversial to hypothesize that grounding lines retreat during warm climate periods and advance during cold climate periods? We are taken aback by the suggestion of this reviewer that we should abandon this hypothesis and are not inclined to follow this suggestion. We believe that the reviewer did not present sufficiently meritorious reasons for us to do that. Scientific peer review should not be used as a backdoor to censorship of scientific ideas.

3. Assumptions of the model: Hodson et al. (2016) demonstrated that sediment transport occurs in a cm's thick layer of deformable till beneath Whillans Ice Stream. This mechanism could result in the transport of old (14C-free) carbon to (and deposition at) sites as ice streams flow over them. This point was overlooked in the manuscript and the response to review.

Author's Note: We apologize for not responding to this point previously. This was our omission. In terms of our mathematical model, the bulk of the C-12 input is incorporated in the time-independent component of equation S9, designated with the symbol 'N_o'. Since C-12 is a stable isotope, it does not experience time-dependent decay the way that C-14 does. Hence, any C-12 incorporated during subglacial erosion may as well be included in the time-independent coefficient 'N_o'. We clarify HERE AND NOW that N_o includes the initial C-12 and

any C-12 that may have been incorporated during subglacial erosion following. We have also added in a statement clarifying this in section 1 of the supplemental.

Given that the reviewer references the result of Hodson et al. (2016), regarding the cm-thick deforming till layer, it is useful to further discuss the potential implication of this constraint for our work. First, this finding of Hodson et al. (2016) does not necessarily apply to '… till beneath Whillans ice stream' in general as it was based on a short sediment core recovered from the bottom of Subglacial Lake Whillans, a setting that may preferentially favor a thin zone of till deformation. It is not clear to us why this reviewer assigns such great significance to this finding because it seems to have limited implication for our results. The subglacial sediment samples that were used for the C-14 measurements come from a range of depths below the core tops, including some samples that come from >1 meter depths. Yet, all these sediment samples contain C-14. Moreover, we have compiled a plot of sample depths versus C-14 concentrations and there is no trend of the latter with depth. Although the reviewer does not make it clear in their comment, it seems that they have in mind the idea that C-14 is abundant at the top of the till and not present or less abundant deeper down. Our data do not support such a contention. Some of the highest C-14 fraction modern values from subglacial samples come from samples that came from deeper parts of the cores, not from core tops. And, as we said, the data does not give any hint about C-14 fraction modern decreasing with depth. Neither is there a significant variability in C-12 concentration with depth or from core to core. For instance, out of 9 sediment samples from beneath Whillans Ice Stream (SLW and WIS localities), seven have TOC of 0.3% and two TOC of 0.4%. This is remarkable homogeneity of C-12 concentration (which is the main component of TOC) with depth and across horizontal distance of about 100 km. So, even if the few-cm-thick deforming layer of Hodson et al. (2016) is causing erosion, the material that is being eroded has similar C-14 and C-12 concentrations as the material that is already incorporated in the thin deforming till layer. Hence, this erosion will not change our modeling results.

As the authors state in their concluding paragraph, this study presents an interesting and new use of old data. However, it is too strong of a claim to state that the modeled chronological constraints for retreat and re-advance can be explained by a climate forcing when viewed through the lens of the full model output. I recommend that acknowledgement of conjecture come before the last paragraph so that the efforts herein are not oversold as a new/alternative explanation for observed Holocene changes. I would suggest that the future version of this manuscript be reframed to focus on the methodological advances so that sufficient explanation for model assumptions, sensitivity, and uncertainty be presented in the main text.

Response to reviewer 2:

Comments on revised Neuhaus et al

The revised paper is substantially improved from the original version I reviewed and I believe that the authors have addressed almost all of my comments. The figures in particular are clearer, the captions explain them better and uncertainty is better represented throughout the figures. As I noted previously I think the paper is a novel contribution to the debate and provides important constraints for understanding Holocene grounding line retreat in the Ross Sea. I also would point out again the importance seeing novel work using innovative approaches to archived samples: this sort of study is very welcome.

In the abstract I think that it would be helpful to include the word 'primarily' or 'dominantly' when referring to climatic controls. The authors have not ruled out GIA processes, only that they suggest climate was dominant. Indeed in the introduction the authors only say that their results are 'consistent with a climatic forcing..'. So last sentence of abstract could read: Based on these results, we propose that the Siple Coast grounding line motions in the mid- to late-Holocene were primarily driven by relatively modest changes in regional climate, rather than by ice sheet dynamics and glacioisostatic rebound, as hypothesized previously (Kingslake et al., 2018).

Author's Note:  We agree with this sentiment that we do not entirely rule out GIA, but rather suggest that our results are more consistent with climatic forcing.  We have thus added in "primarily" to the abstract.

A similar edit is likely to be needed in discussion. This is important and scientifically prudent because the authors do not find a complete match and they cannot rule out some GIA control on readvance.

Author's Note:  We have added in a similar "primarily" to section 4.4 of the discussion.

In my first review I noted that the fields on Fig 9 (now Fig 3) were plotted in Lamb et al (2006) using wt%:wt% not atom:atom (molar) ratios. My point was that there is a conversion between these two units (x1.17) and this needed to be applied if the authors were to plot their data from the atim:atiom measurements. The response claims that Lamb et al uses atom:atom and that therefore the plot in Fig 3 is unmodified. This is not my specialism but I have to note again I have checked and Lamb et al (2006) note very clearly in the text on their p.30 that they use wt% ratios unless stated otherwise (and their field diagram does not state otherwise):
"The weight ratio of organic carbon to total nitrogen (C/N) is normally measured alongside d13C, and can also help to distinguish carbon sources. Occasionally, the weight ratio is converted into an atomic/molar ratio by multiplying by 1.17; however, the ratio used is not always stated. Here, C/N refers to the weight ratio unless stated otherwise." Lamb et al (2006)
I am not a specialist in this field but unless I have missed something, the response implies that the Lamb et al (2006) fields and the data from this study continue to be plotted on different x-scales in Fig 3 and therefore are offset and the plot is in error. This may need someone more expert to take a look or to ask Angela Lamb directly.

Author's Note:  We re-examined what we did to create this figure, and found that the reviewer is correct.  We thought that we had converted from wt:wt to atom:atom, but realized we hadn't.  We have made the necessary corrections.

Line 645: Additional

Author's Note:  Thank you for pointing out this typo.  We have corrected it.

Symbology on Fig 9 - the captions in the revised paper are much richer and more helpful than in the original submission but fig 9 still needs some further explanation of the shaded colour bars at the base of the figure - what does the shading mean, what are the red (?) boxes towards their younger ends and so on. Would be helpful to more clearly and explicitly match the symbology to the plots in Fig 8 e.g. by placing a dot at the optimal GL retreat timing (along with the lines for uncertainty) and a box and whisker for the readvance timing. This might need a small zoom in inset for this part of the figure

Author's Note:  The shaded color bars were just a different representation of the histograms shown in figure 8.  But we like the reviewer's suggestion of changing the symbology to match what is shown in figure 8.  We have now changed the shaded color bars indicating timing of grounding line retreat to dots with error bars (as shown in figure 8), and added the boxes from the figure 8 box and whisker plots of timing of grounding line re-advance.  We have also made sure to use the same colors that were used to indicate the four sites in figure 8.

The discussion of saloon door and swinging gate is now much better: it is nuanced, and points out clearly that the model results presented here cannot distinguish between these two canonical models. The discussion of bathymetric controls is good but I would note, however, that the dependence of retreat on bathymetry has been discussed in many of John Anderson's papers on Antarctic deglaciation for several decades and he should be cited here. The most recent paper, and perhaps most generally applicable is perhaps this one: Anderson,John B. et al.. 2019. Seismic and geomorphic records of Antarctic Ice Sheet evolution in the Ross Sea and controlling factors in its behaviour, Geological Society, London, Special Publications(2019),475(1):223 https://doi.org/10.1144/SP475.5 but there are several earlier examples.

Author's Note:  We agree that John Anderson's work deserves a citation here.  We have therefore added in the citation suggested.

Mike Bentley, July 2021